# Dopaminergic co-transmission with sonic hedgehog inhibits abnormal involuntary movements in models of Parkinson's disease and L-Dopa induced dyskinesia

Lauren Malave[1,2,8], Dustin R. Zuelke[1,3], Santiago Uribe-Cano [1,2], Lev Starikov [1,3,9], Heike Rebholz[1,10,11,12], Eitan Friedman[1,2,3], Chuan Qin[4], Qin Li[4,5], Erwan Bezard [4,5,6,7] & Andreas H. Kottmann [1,2,3✉]

L-Dopa induced dyskinesia (LID) is a debilitating side effect of dopamine replacement therapy for Parkinson's Disease. The mechanistic underpinnings of LID remain obscure. Here we report that diminished sonic hedgehog (Shh) signaling in the basal ganglia caused by the degeneration of midbrain dopamine neurons facilitates the formation and expression of LID. We find that the pharmacological activation of Smoothened, a downstream effector of Shh, attenuates LID in the neurotoxic 6-OHDA- and genetic aphakia mouse models of Parkinson's Disease. Employing conditional genetic loss-of-function approaches, we show that reducing Shh secretion from dopamine neurons or Smoothened activity in cholinergic interneurons promotes LID. Conversely, the selective expression of constitutively active Smoothened in cholinergic interneurons is sufficient to render the sensitized aphakia model of Parkinson's Disease resistant to LID. Furthermore, acute depletion of Shh from dopamine neurons through prolonged optogenetic stimulation in otherwise intact mice and in the absence of L-Dopa produces LID-like involuntary movements. These findings indicate that augmenting Shh signaling in the L-Dopa treated brain may be a promising therapeutic approach for mitigating the dyskinetic side effects of long-term treatment with L-Dopa.

[1] Department of Molecular, Cellular and Biomedical Sciences, CUNY School of Medicine at City College of New York, City University of New York, New York, NY, USA. [2] City University of New York Graduate Center, Neuroscience Collaborative, New York, NY, USA. [3] City University of New York Graduate Center, Molecular, Cellular and Developmental Subprogram, New York, NY, USA. [4] Institute of Laboratory Animal Sciences, China Academy of Medical Sciences, Beijing, People's Republic of China. [5] Motac Neuroscience, Manchester, UK. [6] Universite de Bordeaux, Institut des Maladies Neurodégénératives, UMR 5293, Bordeaux, France. [7] CNRS, Institut des Maladies Neurodégénératives, UMR 5293, Bordeaux, France. [8] Present address: Department of Psychiatry, Columbia University, New York, NY, USA. [9] Present address: Blue Rock Therapeutics, Inc, New York, NY, USA. [10] Present address: GHU Psychiatrie et Neurosciences, Paris, France. [11] Present address: Institut de Psychiatrie et Neurosciences de Paris (IPNP), UMR S1266, INSERM, Universite de Paris, Paris, France. [12] Present address: Center of Neurodegeneration, Danube Private University, Krems, Austria. ✉email: akottmann@med.cuny.edu

Mesencephalic dopamine neuron (DAN) degeneration results in the hallmark motor and cognitive deficits observed in Parkinson's disease[1,2]. Dopamine (DA) replacement therapy with the DA metabolic precursor L-3,4-dihydroxyphenylalanine (L-Dopa) is the gold-standard treatment for Parkinson's disease and attenuates bradykinesia and akinesia[3,4]. Unfortunately, prolonged use of L-Dopa therapy leads to a severely debilitating and uncontrollable side effect called L-Dopa induced dyskinesia (LID) that eventually affects ~90% of all medicated Parkinson's disease patients[5]. These motor impairments impact mobility, produce physical discomfort, negatively impact emotional well-being, and can carry social stigma[6–8].

LID is a medication induced complication that emerges after repeated L-Dopa treatment of Parkinson's disease patients with substantial DAN degeneration but does not present in untreated Parkinson's disease or in treated healthy individuals[2,7,8]. These observations suggest that DAN degeneration alters brain circuitry such that subsequent DA substitution therapy with L-Dopa is insufficient to fully normalize the functional pathology caused by DAN degeneration and instead promotes LID[9]. While the mechanisms through which DAN degeneration enables LID remain unclear, pathophysiological changes in cholinergic inter-neuron (CIN) activity have long been suspected to play a critical role in the formation of LID. For example, it is well established that CIN physiology becomes distorted in complex ways following DAN degeneration with changes in morphology, activity, and the postsynaptic expression of acetylcholine (ACh) receptors[10–18]. Additionally, recent work has demonstrated that treatment with L-Dopa alone is not sufficient to restore the normal properties of CIN in the Parkinson's disease brain, as their disturbed activity and morphology remains or becomes further dysregulated following L-Dopa treatment[19,20]. The complexity of these pathological alterations is highlighted by the fact that both suppressing and ablating CIN[13,17,21], as well as opto-genetic stimulation of CIN[22] and pharmacological augmentation of cholinergic signaling, can attenuate LID in animal models[23].

CIN are direct projection targets of DAN and DA can influence CIN physiology by binding to both inhibitory D2 receptors and facilitatory D5 receptors[24–34]. Dopaminergic control of CIN is further complicated by the fact that DAN also communicate with their targets via the neurotransmitters glutamate[35,36], GABA[37], and the axonaly transported and activity-dependent secreted peptide sonic hedgehog (Shh)[38–42]. Thus, LID might arise in part because L-Dopa treatment alone fails to augment additional modalities of dopaminergic co-transmission. Despite this possibility, whether the loss of dopaminergic co-transmitters following DAN degeneration renders brain circuits susceptible to LID remains to be studied.

Unlike glutamate[43–46] and GABA[37], Shh is expressed by all mesencephalic DAN[38]. Post-synaptically, all CIN express the Shh receptor Patched and its downstream effector Smoothened (Smo)[38] (Fig. 1a). Smo, a $G\alpha_i$-coupled GPCR, is capable of impinging on neuronal intracellular $Ca^{++}$ levels within minutes via G-protein signaling as well as regulating target gene expression within hours via activation of Gli transcription factors[47–49]. DAN degeneration in human and animal models of Parkinson's disease should therefore impact CIN physiology due to a diminishment of both DA and Shh signaling pathway action on CIN (Fig. 1a). The subsequent administration of L-Dopa would then expose CIN to increased levels of DA but relatively diminished levels of Shh (Fig. 1a). We therefore tested the possibility that LID emerge in part because L-Dopa treatment results in an imbalance of Shh and DA signaling onto CIN in the hypodopaminergic brain. More specifically, we hypothesized that augmenting L-Dopa therapy with agonists of Shh signaling could counteract LID induction and expression. Our results confirm these hypotheses and provide guidance toward potential LID therapies.

## Results

**Activating smoothened attenuates dyskinesias in three models of Parkinson's disease**. We first tested whether chronic augmentation of Shh signaling throughout L-Dopa therapy could attenuate the development of dyskinesia in preclinical models of LID with diverse etiologies. Model-specific behavioral tests with predictive validity for examining LID following genetic or neurotoxin-induced degeneration of DAN in rodents and non-human primates have been developed previously[7]. These studies demonstrated that LID develops across different species and forms of induced DAN degeneration in an L-Dopa dose-dependent manner[7]. We quantified LID in these animals by a subjective measure called the abnormal involuntary movement (AIM) scale which captures model-specific features of LID[7,50,51]. In the 6-OHDA rodent model, unilateral striatal injection of the dopaminergic toxin 6-OHDA results in loss of DAN innervation to the ipsilateral dorsal striatum (comparison of changes to DAN striatal innervation and DAN cell counts in the midbrain for all mouse models used in this study are presented in Supplementary Fig. 1) and a rotational bias of locomotion[51] (Supplementary Fig. 2). As previously established[50], daily injections of L-Dopa (5 mg/kg) in these animals led to the appearance of unilateral AIMs (aggregate score of axial, limb, and orofacial contorsions; Fig. 1b, gray bar). We observed that repeated co-injection of L-Dopa with three different doses of Smo agonist SAG[52] resulted in a dose-dependent attenuation of AIMs (Fig. 1b, blue bars). Conversely, repeated co-injection of L-Dopa with Smo antagonist cyclopamine[53] increased AIM scores (Fig. 1b, red bar).

We next tested the effect of Smo stimulation and inhibition on AIMs in the aphakia ($AK^{-/-}$) mouse line. In $AK^{-/-}$ mice, absence of the transcription factor Pitx3 during development results in severe bilateral diminishment of dopaminergic innervation to the dorsal striatum and hypodopaminergia in the adult brain[54,55] (quantified in Supplementary Fig. 1). As has been previously shown, daily injection of $AK^{-/-}$ mice with L-Dopa at 25 mg/kg resulted in presentation of $AK^{-/-}$ model-specific AIMs quantified as the total amount of time that the animal spends sliding 2 or 3 paws along the walls of a glass cylinder while rearing on one leg[56] (Fig. 1c, gray bar and Supplementary Table 1 lists all drug regiments comparatively for all paradigms used in this study). In agreement with our findings in the 6-OHDA LID model, daily co-injections of L-Dopa with SAG (20 mg/kg) resulted in a threefold attenuation of AIMs (Fig. 1c, blue bar), while repeated administration of L-Dopa with cyclopamine (5 mg/kg) increased the duration of AIMs in $AK^{-/-}$ mice (Fig. 1c, red bar) compared to littermate controls.

We next tested whether a single, acute dose of Smo agonist would be effective in attenuating AIMs that were previously established with repeated L-Dopa dosing. A single dose of the Smo agonists purmorphamine[57] (15 mg/kg) or SAG (10 mg/kg) administered alongside the final dose of L-Dopa after repeated daily dosing with L-Dopa alone reduced AIMs in the unilateral 6-OHDA model of LID (Fig. 1d, blue bars). A single dose of SAG (20 mg/kg) in the $AK^{-/-}$ model of LID following repeated L-Dopa administration alone also reduced AIMs (Fig. 1e, blue bar). The ability of SAG to reduce AIMs in $AK^{-/-}$ mice was similar in magnitude to the effect observed following administration of the anti-dyskinetic bench mark drug amantadine (60 mg/kg[58]; Fig. 1e, green bar). Further, the combined application of a single dose SAG (20 mg/kg) and amantadine (60 mg/kg) produced an additive effect on AIM attenuation suggesting these drugs posses distinct mechanisms of action (Fig. 1e, orange bar).

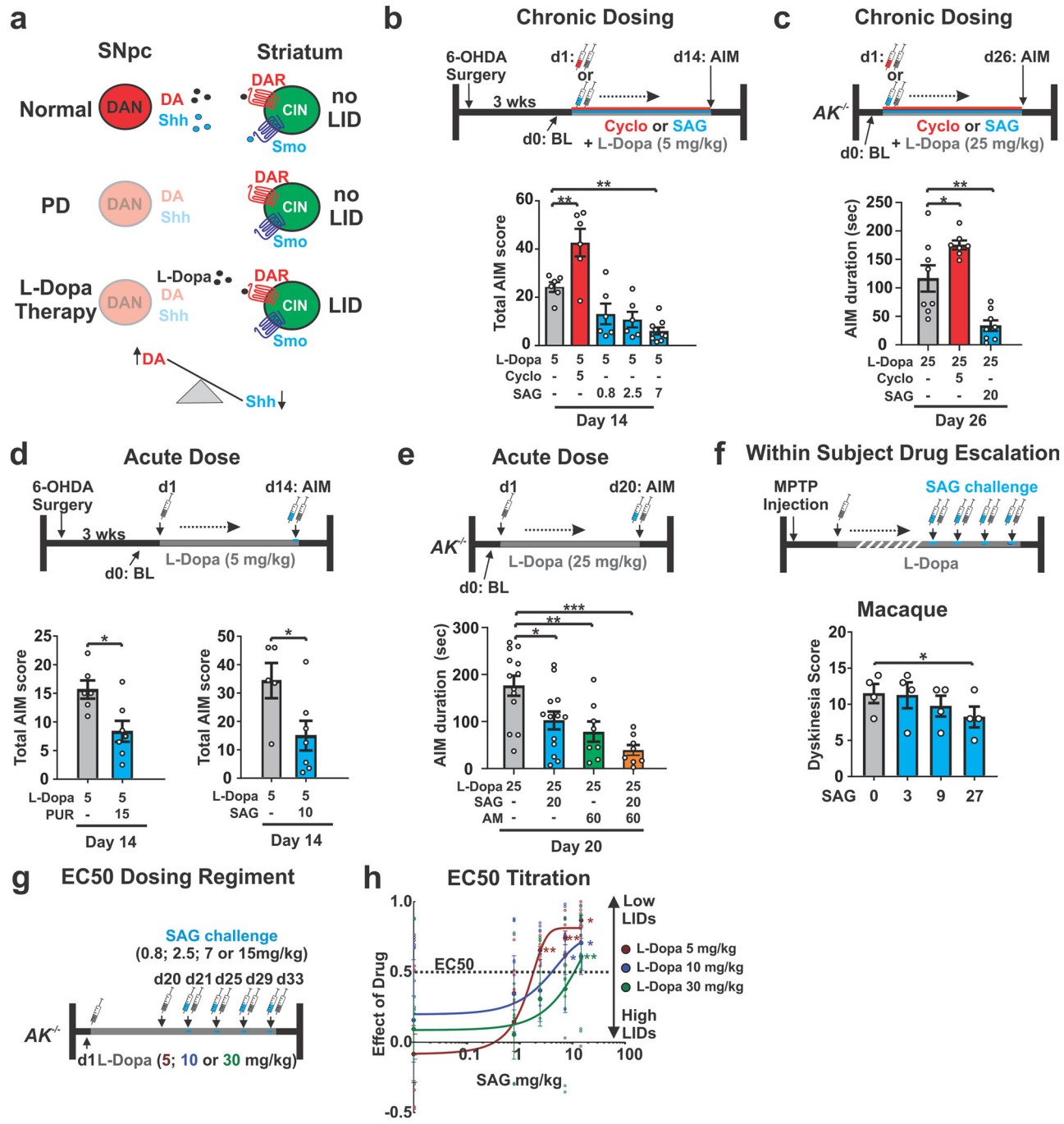

Interestingly, unlike amantadine treatment, neither acute nor chronic dosing with SAG reduced the anti-akinetic benefit of L-Dopa in either of the murine models tested (Supplementary Fig. 3a, b).

In macaques, systemic treatment with the toxin 1-methyl-4-phenyl-1,2,3,6-tetrahydropyridine (MPTP) produces bilateral DAN degeneration leading to Parkinson's disease symptoms. Chronic dosing with L-Dopa reduces parkinsonian disability among MPTP treated macaques but results in the establishment of LID that can be attenuated by amantadine[59]. We utilized subject-specific L-Dopa dosing regimens (18–22 mg/kg; see "Methods") to establish AIMs in MPTP lesioned macaques. We then tested in a within-subject dose escalation experiment whether SAG administration (3; 9; 27 mg/kg) together with L-Dopa resulted in an attenuation of LID (Fig. 1f). We found that

the highest dose of SAG resulted in a modest attenuation of LID (Fig. 1f). As observed in the murine models, SAG treatment did not curtail the anti-akinetic benefits of L-Dopa in parkinsonian monkeys (Supplementary Fig. 3c).

The severity of LID within each animal model is dependent on the dose of L-Dopa administered[49]. We thus investigated whether the dose of Shh needed to attenuate AIMs might scale with the dose of L-Dopa used to induce dyskinesia in the $AK^{-/-}$ and 6-OHDA models. LID were induced in three groups of $AK^{-/-}$ mice injected daily for 20 days with either either 5, 10, or 30 mg/kg L-Dopa (Fig. 1g). In probe trials at days 21, 25, 29, and 33, the dose-dependent degree of LID attenuation by SAG was tested using a within-subject escalation strategy (0.8; 2.5; 7.5; and 15 mg/kg) in each L-Dopa dose group (5, 10, or 30 mg/kg; Fig. 1g). While the lowest dose of SAG did not have an effect on AIMs in any of the

**Fig. 1 Pharmachological manipulation of Smo bidirectionally modulates LID in neurotoxic and genetic models of Parkinson's Disease. a** DA substitution therapy in the Parkinson's disease brain without augmentation of $Shh_{DAN}$ signaling likely causes an imbalance between DA (high) and Shh (low) signaling onto CIN. **b** 6-OHDA mice dosed daily for 14 days with L-Dopa (5 mg/kg) expressed robust LID gray bar). Daily co-injection of L-Dopa with the Smo antagonist cyclopamine (cyclo; 5 mg/kg, red bar) increased LID while daily co-injection of L-Dopa with the Smo agonist SAG (0.8, 2.5, or 7 mg/kg; blue bars) decreased LID ($n = 6$–9; RM two-way ANOVA: $F_{(4,28)} = 12.24$, $p < 0.0001$; post hoc Bonferroni's test: *$p < 0.05$, **$p < 0.01$ treatment vs. control). **c** Compared to AIM duration observed in $AK^{-/-}$ mice dosed daily for 26 days with L-Dopa (25 mg/kg) and vehicle, daily co-injection of L-Dopa with cyclopamine (5 mg/kg; red bar) extended AIMs duration while daily co-injection of L-Dopa with Smo agonist SAG (20 mg/kg; blue bar) decreased AIMs duration ($n = 7$–8; one-way ANOVA: $F_{(2,20)} = 20.46$, $p < 0.0001$; post hoc Bonferroni's test: *$p < 0.05$, **$p < 0.01$ treatment vs. control). **d** A single dose of the Smo agonist purmorphamine (PUR; 15 mg/kg; blue bar, left) or SAG (10 mg/kg; blue bar; right) attenuated LID established previously by 14 days of daily L-Dopa (5 mg/kg) dosing ($n = 7$ per treatment; unpaired two-tailed Student's $t$ test *$p < 0.05$ treatment vs. control). **e** Expression of AIMs established through daily L-Dopa (25 mg/kg) dosing in $AK^{-/-}$ mice was reduced after a single-dose treatment of either SAG (20 mg/kg, blue bar), amantadine (AM, 60 mg/kg, green bar), or SAG combined with AM (orange bar) compared to vehicle-treated controls (gray bar) ($n = 8$–13 per treatment; one-way ANOVA, $F_{(4,50)} = 9.79$, $p < 0.0001$; post hoc Bonferroni's test: *$p < 0.05$, **$p < 0.01$, ***$p < 0.001$ treatment vs. control). **f** Dyskinesia score in dyskinetic Macaques was reduced in response to a within-subject drug escalation of SAG at the highest dose (0, 3, 9, and 27 mg/kg; $n = 4$ (Friedman's nonparametric RM one-way ANOVA, $F = 10.89$, $p = 0.0020$, followed by Dunn's post hoc test, $p = 0.0185$). **g** LID were induced in three groups of $AK^{-/-}$ mice injected daily for 20 days with either 5, 10, or 30 mg/kg L-Dopa. In probe trials at days 21, 25, 29, and 33, the dose-dependent degree of LID attenuation by SAG was tested using a within-subject escalation strategy including three-day SAG washouts (L-Dopa only) inbetween probe trials. SAG was serially administered across the probe trials at four different concentrations (0.8; 2.5; 7.5; and 15 mg/kg) in each L-Dopa dose group (5, 10, or 30 mg/kg). **h** In each L-Dopa concentration group, the attenuation of LID by SAG was SAG dose-dependent. Half-maximal (EC50) inhibition of LID reveals a positive correlation between the SAG concentration needed for LID attenuation and the L-Dopa dose used to induce LID. Stars represent significance from baseline AIMs when only L-Dopa and vehicle were administered ($n = 8$ for each drug condition; unpaired two-tailed Student's $t$ test *$p < 0.05$, **$p < 0.01$. n.s. indicates $p > 0.05$. Graph is plotted as mean ± SEM).

L-Dopa dosing groups, the higher doses resulted in a dose-dependent attenuation of AIMs in each of the L-Dopa groups. This study revealed that the dose of SAG needed to attenuate LID by 50% in each of the L-Dopa groups scaled with the dose of L-Dopa used to induce dyskinesia (Fig. 1h). Concordant with the L-Dopa to SAG dose relationship in $AK^{-/-}$ mice, we observed in the 6-OHDA paradigm that greater doses of SAG (20 mg/kg) were required to attenuate AIMs after L-Dopa was ramped up from 5 to 20 mg/kg. Conversely, AIMs produced by lower L-Dopa doses in the same animals (5 mg/kg) could be attenuated by treatment with lower doses of SAG (10 mg/kg) (Supplementary Fig. 4). Finally, we found that AIMs could be repeatedly attenuated or reinstated, respectively, by dosing the same animals sequentially with or without SAG in addition to L-Dopa (Supplementary Fig. 4).

**Smo activity modulates the MAP kinase pathway in CIN.** Phosphorylation of extracellular signal-regulated protein kinase (Erk$^{1/2}$) in the striatum is a cell physiological marker for LID[60]. During early LID, pErk$^{1/2}$ (phospho-Thr202/Tyr204–ERK$^{1/2}$) is observed in a dispersed pattern within the striatum among primarily medium spiny projection neurons (MSN) expressing the DA D1 receptor[60]. However, dosing with L-Dopa progressively increases pErk$^{1/2}$ prevalence among CIN in the dorsolateral striatum (DLS) in correlation with increased LID severity[13]. Conversely, pharmacological inhibition of Erk phosphorylation attenuates LID[13]. We therefore examined how the relative degree of Erk phosphorylation in 6-OHDA and $AK^{-/-}$ mice is affected by manipulations of Shh signaling. We found that daily co-injections of L-Dopa with either the antagonist cyclopamine (Fig. 2a, b, red bars) or the agonist SAG (Fig. 2a, b, blue bars) of Smo resulted in a respective increase or decrease in the fraction of pErk$^{1/2}$-positive CIN of the DLS 30 min after the final co-administration. The fraction of pErk$^{1/2}$-positive CIN in the dorsomedial striatum (DMS) of the same animals was not affected (Fig. 2b). Similarly, a single dose of the Smo agonists SAG or purmorphamine co-injected with the final daily dose of L-Dopa reduced the prevalence of pErk$^{1/2}$ positive CIN in the DLS of 6-OHDA and $AK^{-/-}$ mice 30 min after drug administration (Fig. 2c). As observed for the chronic dosing, the acute administration of purmorphamine or SAG had no effect on Erk phosphorylation in the DMS (Fig. 2c). Parvalbumin expressing fast spiking interneurons of the striatum are also sensitive to Shh

signaling from DAN and express Ptc1 and Smo[38]. However, Smo agonist treatment did not alter the phosphorylation status of pERK in fast spiking interneurons in the striatum of L-Dopa treated 6-OHDA or $AK^{-/-}$ animals (Supplementary Fig. 5).

**Smo modulates AIMs via a mechanism upstream of cholinergic influence on striatal output.** We next investigated whether the action of Smo relevant to LID modulation was upstream of DA receptor 1 expressing MSNs whose aberrant striatal output in models of LID is known to result in AIMs[7,38]. It was previously shown that muscarinic ACh receptor M4 positive allosteric modulators (M4-PAM) restores long-term depression in DA receptor 1 expressing MSNs of the striatal direct output pathway and attenuates LID[23,61]. Muscarinic ACh receptor M4 also acts as an inhibitory autoreceptor on CIN. Thus, our observation that the Smo antagonist cyclopamine facilitates LID formation allowed us to test if Smo modulation of LID antecedes M4-PAM mediated LID attenuation (Fig. 3a). We induced AIMs in $AK^{-/-}$ mice by injecting L-Dopa (25 mg/kg) paired with either vehicle or cyclopamine (5 mg/kg) for 20 days. Mice were challenged with a single dose of M4-PAM VU0467154 (10 mg/kg) alone or in combination with cyclopamine (5 mg/kg) on the final day (Fig. 3b). As before, we found that cyclopamine significantly increased the severity of AIMs compared to animals treated with L-Dopa alone (Fig. 3c, red bar vs. white bar). On the contrary, M4-PAM treatment blocked the aggravating effect of cyclopamine on AIMs and produced levels similar to those observed in M4-PAM and L-Dopa treated $AK^{-/-}$ control mice (gray bars; Fig. 3c). These findings are consistent with the possibility that Smo-mediated modulation of AIMs occurs upstream of M4-PAM dependent attenuation of AIMs, and therefore presynaptic to cholinergic inputs onto DR1expressing MSN.

**Conditional ablation of Smo from CIN facilitates LID while expression of constitutively active SmoM2 in CIN blocks LID in the $AK^{-/-}$ model.** If Smo expressed by CIN is in fact the relevant target of Smo antagonists and agonists that respectively promote or attenuate LID and Erk activation, then the conditional genetic loss or gain of function of Smo in CIN should phenocopy the pharmacological outcomes. To begin testing this, we produced mice with conditional ablation of Smo from CIN ($Smo_{CIN}^{-/-}$, Fig. 4a; generation of all conditional gene ablation and gain of

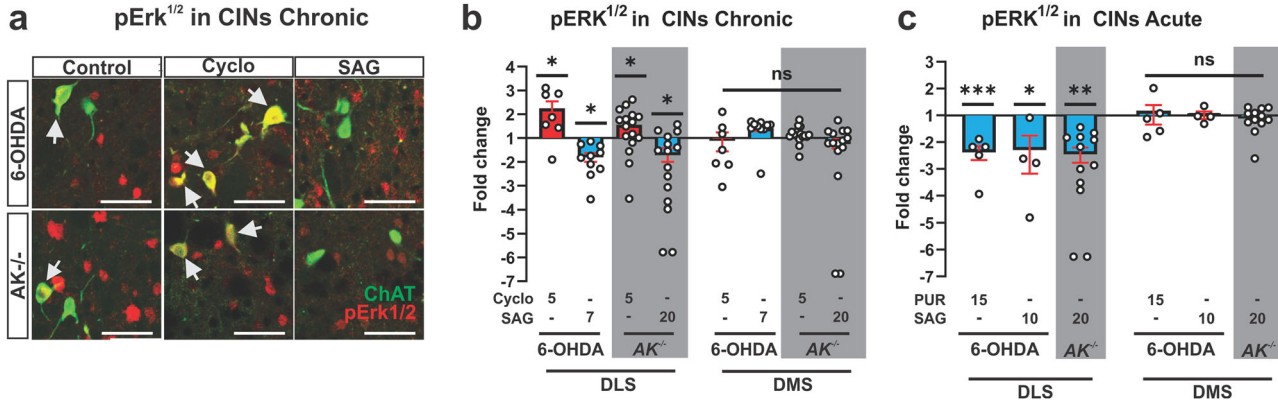

**Fig. 2 Smo activity modulates the MAP Kinase pathway in CIN. a** Representative images from DLS of 6-OHDA or $AK^{-/-}$ mice showing co-localization (white arrows) of the cytohistological LID marker pErk$^{1/2}$ (red) with the CIN marker ChAT (green). Images were taken from DLS of animals whose AIMs were quantified in **b** and **c** of Fig. 1 and that underwent repeated co-administration of L-Dopa (5 mg/kg for 6-OHDA or 25 mg/kg for $AK^{-/-}$ animals) with vehicle (control), cyclopamine (5 mg/kg), or SAG (7 mg/kg for 6-OHDA mice or 20 mg/kg for $AK^{-/-}$ mice). Scale bar = 50 μm. **b** Quantification of the prevalence of pErk$^{1/2}$-positive CIN expressed as fold change over vehicle in the DLS and DMS. cyclopamine treatment increased pErk$^{1/2}$ prevalence and SAG treatment decreased pErk$^{1/2}$ prevalence in the DLS with no changes in the DMS ($n = 7$–15 per condition; three sections each; ~36 CIN per section, postmortem analysis of animals quantified in Fig. 1b, c; unpaired two-tailed Student's $t$ test: *$p < 0.05$ treatment vs. vehicle. n.s. indicates $p > 0.05$). **c** Quantification of the prevalence of pErk$^{1/2}$-positive CIN in L-Dopa treated 6-OHDA (5 mg/kg L-Dopa for 14 days) or $AK^{-/-}$ (25 mg/kg L-Dopa for 20 days) animals given a single dose of PUR (15 mg/kg) or SAG (10 mg/kg for 6-OHDA animals, 20 mg/kg for $AK^{-/-}$ animals). Results are expressed as fold change in pErk$^{1/2}$ exspression over vehicle controls in the DLS and DMS. Smo agonist treatment significantly reduced the prevalence of pErk$^{1/2}$-positive CIN in the DLS but not in the DMS ($n = 5$–13 per condition; three sections each; ~36 CIN per section, postmortem analysis of animals whose AIMs were quantified in Fig. 1d, e; unpaired two-tailed Student's $t$ test: *$p < 0.01$, ***$p < 0.001$, for treatment vs. vehicle. n.s. indicates $p > 0.05$). All bar graphs are plotted as mean ± SEM.

function mouse lines described in methods; allelic configuration of experimental and control animals of all recombinant mouse lines used in this study is listed in Supplementary Table 2; all drug dosing regimens were started around 8 weeks of age and all final behavioral and histological analysis was carried out between 12 and 14 weeks of age). These mice showed no evidence of dopaminergic fiber density atrophy in the striatum and possessed a full complement of DAN in the midbrain at 3 months of age (quantified in Supplementary Fig. 1). We examined whether these mice would develop LID in response to daily L-Dopa dosing (25 mg/kg, 20 days) in the absence of any other manipulation (Fig. 4b). When scored for AIMs at 3 months of age by the same methodology used for the $AK^{-/-}$ paradigm, we found a significant increase in AIMs among homozygous mutants ($Smo_{CIN}^{-/-}$) compared to heterozygous controls ($Smo_{CIN}^{-/+}$; Fig. 4c; white bar compared to red bar). $Smo_{CIN}^{-/+}$ also revealed a significant increase in AIMs compared to baseline measures (day 1), indicating a Smo gene dose-dependent formation of LID (Fig. 4c). Concordant with the behavior, $Smo_{CIN}^{-/-}$ mice also exhibited a corresponding increase in the fraction of pErk$^{1/2}$-positive CIN of the DLS compared to $Smo_{CIN}^{-/+}$ (Fig. 4d). In order to test whether the pharmacological activation of Smo specifically residing on CIN could have been responsible for the attenuation of LID we expressed a constitutively active form of Smo, the oncogene SmoM2[62], selectively in CIN of $AK^{-/-}$ mice ($SmoM2_{CIN}^{+/-}$; $AK^{-/-}$, Fig. 4e and Supplementary Fig. 6). Comparative quantification of striatal DAN fiber density and numbers of DAN in the midbrain at 3 months of age revealed that selective expression of SmoM2 in CIN did not alter the severe diminishment of DAN observed in the parental $AK^{-/-}$ mouse line (quantified in Supplementary Fig. 1). SmoM2 expression on CIN of $AK^{-/-}$ animals ($SmoM2_{CIN}^{+/-}$; $AK^{-/-}$) blocked the formation and progressive intensification of AIMs, which daily dosing with L-Dopa at 25 mg/kg readily produced in $AK^{-/-}$ ChAT-cre littermate controls (Fig. 4g), a dose that resulted in LID in AK−/− line (Fig. 1c). Concordant with the resistance to LID formation, $SmoM2_{CIN}^{+/-}$; $AK^{-/-}$ mice displayed a corresponding decrease in the prevalence of pErk$^{1/2}$-positive CIN of

the DLS compared to $AK^{-/-}$ control mice (Fig. 4h). Together, the genetic Smo conditional loss and gain of function studies indicate that Smo activation on CIN is necessary and sufficient to curtail LID.

**Shh originating from DAN suppresses L-Dopa induced dyskinesia.** Given that DAN of the Substantia Nigra pars compacta (SNpc) express Shh ($Shh_{DAN}$) throughout life, target CIN of the DLS, and degenerate in Parkinson's disease, DAN are a likely physiological source of Shh that could impinge on Smo activity in CIN[38,63,64]. We therefore tested whether mice with a conditional ablation of Shh from DAN ($Shh_{DAN}^{-/-}$) would develop LID (Fig. 5a). $Shh_{DAN}^{-/-}$ mice go through a phase of exuberant DAN fiber sprouting resulting in a ~38 ± 6.8% increase in dopaminergic fiber density in the striatum while harboring a normal number of DAN within the midbrain as young adults (quantified in[38] and in Supplementary Fig. 1). Despite this increased striatal innervation, daily L-Dopa dosing of these animals at 10 mg/kg led to progressive LID expression in $Shh_{DAN}^{-/-}$ mice but not heterozygous littermate controls. LID in $Shh_{DAN}^{-/-}$ mice could be further exacerbated by increasing the L-Dopa dose to 25 mg/kg (Fig. 5b, c). To determine if postsynaptic activation of Smo by SAG could rescue the LID phenotype of $Shh_{DAN}^{-/-}$ mice, we injected $Shh_{DAN}^{-/-}$ and heterozygous control mice with 25 mg/kg L-Dopa daily for 20 days, followed by either 10 or 20 mg/kg SAG alongside the final L-Dopa dose (Fig. 5d). As before, we found that L-Dopa dosing resulted in AIMs among $Shh_{DAN}^{-/-}$ mice but not heterozygous littermate controls (Fig. 5e). However, a single dose of SAG combined with the last dose of L-Dopa was able to significantly attenuate LID at both SAG concentrations in $Shh_{DAN}^{-/-}$ mice, with the higher concentration having a greater effect (Fig. 5e). Concordant with the behavioral phenotype, L-Dopa dosing in $Shh_{DAN}^{-/-}$ mice resulted in an increase in the fraction of pErk$^{1/2}$-positive CIN of the DLS compared to controls, which acute SAG co-injection with L-Dopa normalized to control levels (Fig. 5f, g). These findings indicate a LID-preventative

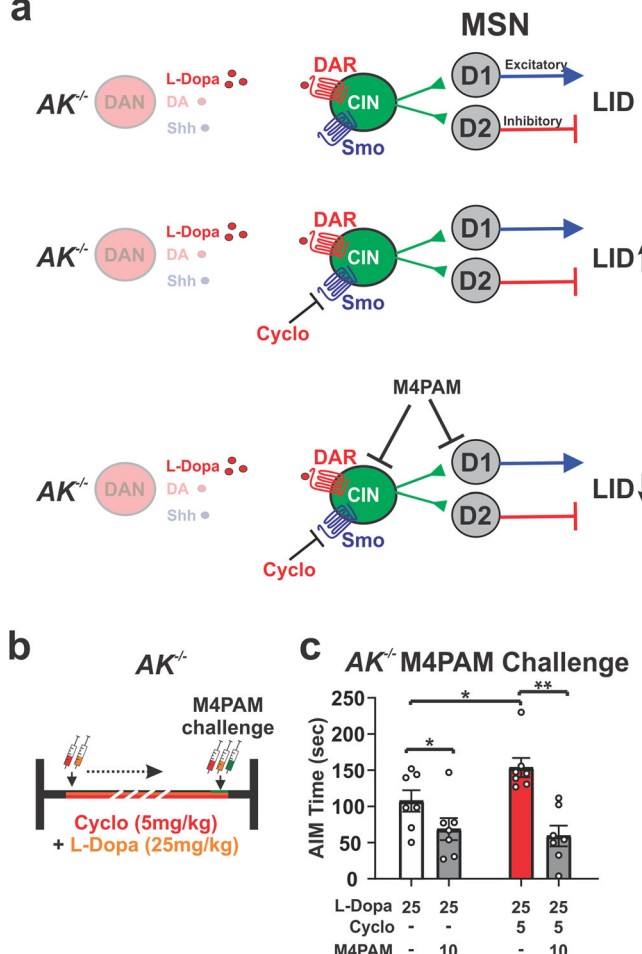

**Fig. 3 ShhDAN to SmoCIN signaling attenuates AIMs via a mechanism upstream of cholinergic influence on striatal output pathways. a** The exacerbation of LID by cyclopamine (Fig. 1) and the well studied attenuation of LID by M4-PAM allowed us to test whether Smo-mediated modulation of AIMs in $AK^{-/-}$ mice occurs upstream or downstream of cholinergic action on DR1-MSNs. **b** Dosing schedule of $AK^{-/-}$ mice with L-Dopa (25 mg/kg) and vehicle or cyclopamine (5 mg/kg) for 20 days. Mice were challenged with a single dose of M4-PAM VU0467154 (10 mg/kg) alone or in combination with cyclopamine (5 mg/kg) on the final day. **c** M4-PAM attenuated cyclopamine-caused facilitation of AIMs (red bar) in $AK^{-/-}$ animals down to the same levels achieved by M4-PAMs in $AK^{-/-}$ mice treated with L-Dopa alone (gray bars; $n = 7$; paired two-tailed Student's t test $p < 0.05$, $*p < 0.05$, $**p < 0.01$ veh vs. M4-PAM; unpaired two-tailed Student's t test $p < 0.05$, $*p < 0.05$ vehicle vs. cyclopamine). Bar graph is plotted as mean ± SEM.

mechanism by which CIN physiology is impacted by Shh provided by DAN. Thus, the suppression of LID by ShhDAN via SmoCIN occurs likely antecedent to LID inhibition by muscarinic signaling on DA receptor 1 expressing MSNs via M4-PAM[23] (Fig. 3), or by nicotinic receptor desensitization proposed to occur upon optogenetic CIN activation[22].

**Depletion of Shh from DAN via prolonged stimulation facilitates abnormal involuntary movements**. Our observations indicate that an imbalance of low Shh signaling due to DAN degeneration in conjunction with boosted dopaminergic signaling onto CIN due to L-Dopa facilitates LID. In order to resolve whether this imbalance between DA and Shh signaling onto CIN

is in fact sufficient to induce AIMs independent of L-Dopa and possible ectopic dopaminergic signaling caused by L-Dopa, we sought to develop a paradigm in which we can acutely deplete Shh from DAN.

High-frequency stimulation is capable of triggering Shh release from neurons[42]. However, prolonged stimulation of neurons can produce an acute exhaustion of peptide stores and deplete peptide signaling over time in a manner not observed with small neurotransmitters[65–67]. We therefore explored whether imposing prolonged DAN burst activity through optogenetic stimulation would result in a progressive reduction of ShhDAN signaling. Forced, unilateral pulsatile burst firing of DAN[42] (Fig. 6a) resulted in optical stimulation-dependent rotational activity that intensified over the course of an hour of stimulation, and thus indicated a progressive change in DAN signaling across the session (Fig. 6b). These effects were observed without detectable change in postmortem dopaminergic fiber density within the striatum (Supplementary Fig. 7). To determine whether the intensification of rotational activity was in part elicited by the exhaustion of Shh signaling, we examined the effects of Smo pharmacology on rotational behavior. Smo inhibition via cyclopamine injection (5 mg/kg) 30 min prior to onset of optogenetic stimulation elicited a maximal rotational response immediately at the beginning of stimulation, which otherwise was only observed after 1 h of prolonged stimulation in carrier injected control mice (Fig. 6c). Conversely, injection of the Smo agonist SAG (20 mg/kg) 30 min prior to onset of optogenetic stimulation blocked the gradual increase in rotational behavior seen in control mice (Fig. 6d). Thus, the bidirectional effects of Smo agonist and antagonist treatment on the progressive intensification of circling behavior indicated that optogenetic stimulation of DAN produced gradual diminishment of ShhDAN signaling. Given the differences in susceptibility to transmitter exhaustion between DA and Shh, these results suggest that prolonged forced stimulation of DAN creates an imbalance of relatively high DA and diminished ShhDAN signaling onto CIN, similar to the imbalance of DA and Shh signaling that facilitates the expression of AIMs in the neurotoxic and genetic paradigms described above (Figs. 1–5).

Concordant, we found that LID-like AIMs, which we named optical-induced dyskinesias (OID), emerged toward the end of stimulation on day 1 and were time-locked with laser onset (Fig. 6e; AIMs summed in 10 min bins, Supplementary Movie 1). These dyskinetic movements intensified in response to daily 1-h stimulation sessions for 10 days (Fig. 6e and Supplementary Movie 2). A single dose of SAG administered prior to optogenetic stimulation on day 11 was sufficient to significantly attenuate OID (Fig. 6e). Quantification of AIM subtypes ("Methods") revealed that SAG treatment reduced limb dyskinesia to levels observed in unstimulated controls while axial dyskinesia was uneffected (Fig. 6f). These results indicate that LID-like AIMs can emerge within minutes of pulsatile DAN burst firing concomitant with the exhaustion of ShhDAN release.

**Shh signaling from DAN increases phosphorylation of the neuronal activity marker p-rpS6 in CIN via Smo activation**. The rapid modulation of LID-like behavior following changes in Shh signaling suggested that ShhDAN might impinge on CIN neuronal activity via Smo. In CIN, levels of the ribosomal activity marker p-rpS6$^{240/244}$ have been shown to correlate positively with CIN activity under basal and stimulated conditions[68–70]. Therefore, we next examined whether ShhDAN/SmoCIN signaling can affect p-rpS6$^{240/244}$ levels averaged across CIN of the DSL.

In agreement with the previous finding that muscarinic autoreceptor M2 expression was increased and, conversely,

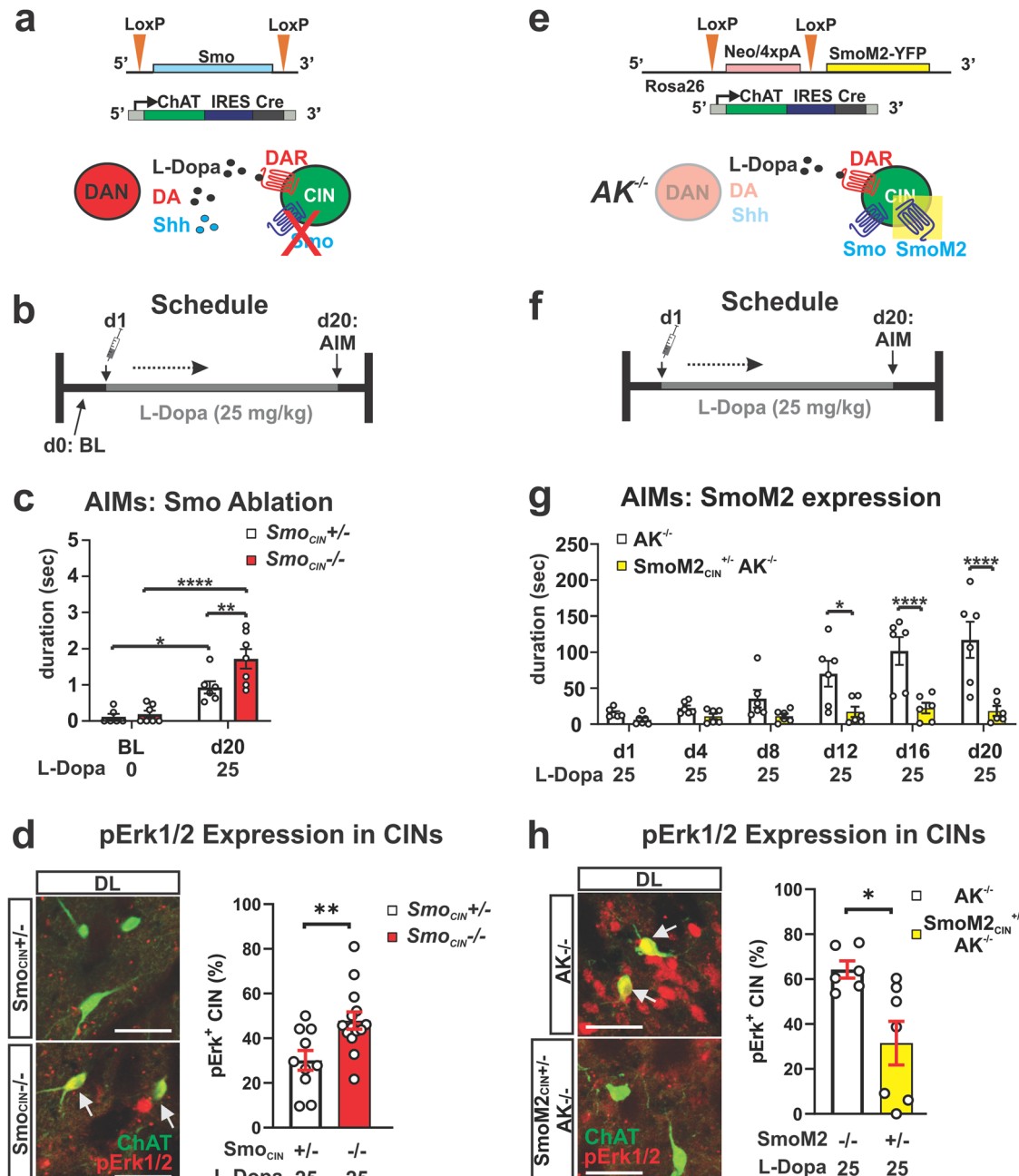

**Fig. 4 Conditional genetic ablation of Smo from CIN facilitates LID while conditional expression of constitutively active SmoM2 in CIN inhibits LID. a** Smo was selectively ablated from CIN by ChAT-IRESCre, which renders CIN insensitive to endogenous Shh signaling. **b** L-Dopa dosing schedule for the Smo loss-of-function study. **c** AIMs are increased in $Smo_{CIN}{}^{-/-}$ mice compared to $Smo_{CIN}{}^{+/-}$ heterozygous control littermates in response to daily injection of 25 mg/kg L-Dopa for 20 days ($n = 6–7$ per genotype; RM two-way ANOVA day × genotype effect: $F_{(1,11)}$ 4.95, $p = 0.048$. Post hoc Bonferroni's test: **$p < 0.01$ control vs. mutant on day 20; day effect: $F_{(1,11)} = 52.61$, $p < 0.0001$; *$p < 0.05$ control BL vs. day 20 and ****$p < 0.0001$ mutant BL vs. day 20). **d** Representative images of pErk$^{1/2}$ (red) co-localization (arrow) with CIN (ChAT, green). Quantification of pErk$^{1/2}$ prevalence in the posterior DLS of $Smo_{CIN}{}^{-/-}$ and $Smo_{CIN}{}^{+/-}$ heterozygous control littermates was performed following daily dosing with L-Dopa (25 mg/kg) for 20 days ($n = 10–14$ per genotype; three sections each; 35–38 CIN per section). Unpaired two-tailed Student's $t$ test **$p < 0.01$ control vs. mutant. n.s. indicates $p > 0.05$. Scale bar = 50 μm. **e** Constitutively active SmoM2 was selectively expressed in CIN of $AK^{-/-}$ mice by ChAT-IRESCre, thus activating Smo signaling in CIN independent of endogenous Shh availilbility. **f** L-Dopa dosing schedule for the SmoM2 gain-of-function study. **g** AIMs formation in response to daily injection of 25 mg/kg L-Dopa is suppressed in $SmoM2_{CIN}{}^{+/-}AK^{-/-}$ mice compared to $AK^{-/-}{}_{ChAT-Cre}{}^{+/-}$ littermate controls ($n = 6$ per genotype; RM two-way ANOVA time × genotype effect: $F_{(5,50)} = 8.16$, $p = <0.0001$. Post hoc Bonferroni's test: *$p < 0.05$, ****$p < 0.0001$ control vs. mutant). **h** Representative images of pErk$^{1/2}$ (red) co-localization (arrows) with CIN (ChAT, green). Quantification of pErk$^{1/2}$ prevalence in the posterior DLS and DMS of $SmoM2_{CIN}{}^{+/-}AK^{-/-}$and $AK^{-/-}{}_{ChAT-Cre}{}^{+/-}$ littermate controls was performed following daily dosing with L-Dopa (25 mg/kg) for 20 days ($n = 6–7$ per genotype; three sections each; 35–38 CIN per section). Unpaired two-tailed Student's $t$ test *$p < 0.05$ control vs. mutant. n.s. indicates $p > 0.05$. All bar graphs are plotted as mean ± SEM. Scale bar = 50 μm.

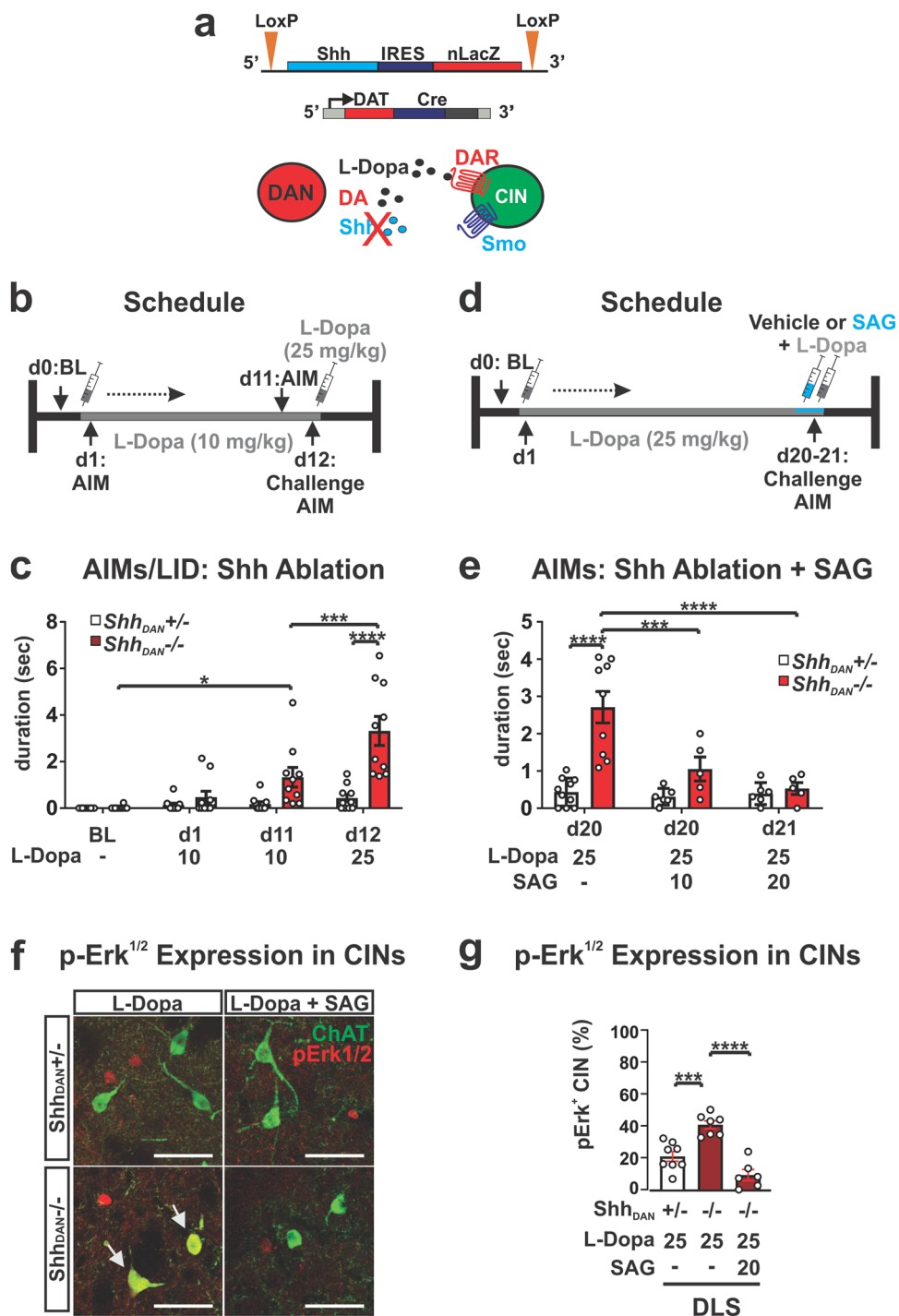

cholinergic tone reduced in the striatum of $Shh_{DAN}{}^{-/-}$ mice[38], we found reduced p-rpS6[240/244] levels in CIN of $Shh_{DAN}{}^{-/-}$ mice compared to controls (Fig. 7a, b). A single dose of SAG administered 30 min prior to analysis, restored p-rpS6[240/244] levels in $Shh_{DAN}{}^{-/-}$ mice (Fig. 7a, b). Similarly, the selective ablation of Smo from CIN in $Smo_{ChAT-Cre}{}^{-/-}$ animals phenocopied the reduction in p-rpS6[240/244] levels among CIN of $Shh_{DAN}{}^{-/-}$ mice (Fig. 7c, d). However, in these animals, SAG failed to restore CIN p-rpS6[240/244] levels indicating that SAG dependent alterations of CIN physiology are direct and that $Shh_{DAN}$ normalizes p-rpS6[240/244] levels through Smo activation on CIN (Fig. 7c, d). Interestingly, mice expressing constitutively active SmoM2 in CIN

($SmoM2_{ChAT-Cre}{}^{+/-}$ mice) did not exhibit further increases in average p-rpS6[240/244] levels among CIN, suggesting that basal levels of Shh signaling produce a maximal effect on p-rpS6[240/244] levels in CIN (Fig. 7c, d). Finally, animals that underwent 1 h of DAN optogenetic stimulation in the Shh acute depletion paradigm showed decreased p-rpS6[240/244] levels in CIN of the stimulated ipsilateral DLS, but not contralateral unstimulated side (Fig. 7e, f). SAG administration prior to onset of optogenetic stimulation blocked this decrease of p-rpS6[240/244] levels in CIN (Fig. 7e, f). Together, these observations indicate that $Shh_{DAN}$ to $Smo_{CIN}$ signaling contributes to cholinergic activity in a direct, positive and acute manner as measured by p-rpS6[240/244].

**Fig. 5 Conditional genetic ablation of Shh from DAN facilitates AIMs which can be attenuated by agonists of Smo. a** Shh was selectively ablated from DAN by Dat-Cre which reduces Shh/Smo signaling in the striatum. **b** L-Dopa dosing schedule for the $Shh_{DAN}$ loss-of-function study. **c** Daily injection of 10 mg/kg L-Dopa for 11 days in $Shh_{DAN}^{-/-}$ mice compared to controls elicited AIMs which were further increased in response to a 25 mg/kg L-Dopa challenge dose ($n = 10$ per genotype; RM two-way ANOVA day × genotype: $F_{(2,36)} = 4.53$, $p = 0.018$. Post hoc Bonferroni's test: *$p < 0.05$ for BL vs. day 11); treatment effect: $F_{(1,18)} = 16.23$, $p = 0.001$; post hoc Bonferroni's test: ***$p < 0.001$ day 11 vs. day 12; genotype effect: $F_{(1,18)} = 18.36$, $p = 0.0004$; post hoc Bonferroni's test: ****$p < 0.0001$ control vs. mutant on day 12). **d** L-Dopa and SAG dosing schedule for the pharmacological complementation of the LID phenotype seen in $Shh_{DAN}^{-/-}$ mutants. **e** A single injection of SAG (10 mg/kg or 20 mg/kg) attenuated AIMs that were established by daily injections of L-Dopa (25 mg/kg) for 20 days in $Shh_{DAN}^{-/-}$ mice. No effect was observed in $Shh_{DAN}^{+/-}$ heterozygous littermate control mice ($n = 5$–10; two-way ANOVA genotype effect: $F_{(1,35)} = 21.36$, $p = 0.0002$. Post hoc Bonferroni's test: ****$p < 0.0001$ control vs. mutant on day 20; genotype × treatment effect: $F_{(2,35)} = 9.15$, $= 0.001$. Post hoc Bonferroni's test: ***$p < 0.001$, ****$p < 0.0001$ mutant with and without SAG on day 20 and day 21). **f** Images of pErk$^{1/2}$ (red) co-localization (arrow) with ChAT (green) in the DLS of $Shh_{DAN}^{-/-}$ mutants and $Shh_{DAN}^{-/+}$ heterozygous littermate controls following chronic daily dosing with L-Dopa alone or L-Dopa with SAG (postmortem analysis of animals shown in **c**); scale bar = 50 µm. **g** L-Dopa increased the prevalence of pErk$^{1/2}$-positive CIN in the DLS of $Shh_{DAN}^{-/-}$ mice. This increase could be attenuated through co-injection of SAG with L-Dopa ($n = 8$–12, 35–38 CIN per genotype and drug condition; two-way ANOVA Treatment effect: $F_{(5,30)} = 14.58$, $p < 0.0001$; post hoc Bonferroni's test: ****$p < 0.0001$ mutant L-Dopa vs. mutant L-Dopa + SAG, ***$p < 0.001$ heterozygous L-Dopa vs. mutant L-Dopa; **$p < 0.01$ mutant L-Dopa DLS vs. mutant L-Dopa DMS). All bar graphs are plotted as mean ± SEM.

## Discussion

In Parkinson's disease patients, LID develops in response to pulsatile administration of L-Dopa. Here, we present several lines of evidence indicating that the increase in DA signaling produced by L-Dopa acts in concert with reduced Shh signaling following DAN degeneration to facilitate the formation and expression of LID: (1) in the 6-OHDA mouse-, MPTP macaque-, and $AK^{-/-}$ genetic mouse models of DAN loss, agonists of the Shh signaling effector Smo attenuate LID formation and expression (Fig. 8a); (2) conditional presynaptic ablation of Shh from DAN or postsynaptic Smo ablation from CIN facilitate LID (Fig. 8b); (3) CIN-specific expression of constitutively active Smo, SmoM2, blocks LID formation in the $AK^{-/-}$ mouse model of LID (Fig. 8c); (4) acute depletion of Shh$_{DAN}$ by prolonged optogenetic stimulation of DAN in otherwise intact mice results in LID-like AIMs (Fig. 8d).

By identifying the commonality among these diverse paradigms which differ greatly in their etiology, we were able to pinpoint that the relative imbalance between Shh (low) and DA (high) signaling onto CIN is a critical facilitator of LID and a determinant of the severity of LID. We further demonstrate that these processes can occur independent of DAN integrity and striatal innervation so long as DA and Shh signaling onto CIN remain unbalanced. Additionally, we find that augmenting Shh signaling alongside L-Dopa administration attenuates LID by re-balancing Shh and DA levels that act on CIN. . Thus, our data indicate that signaling from Shh$_{DAN}$ to Smo$_{CIN}$ prevents LID formation and expression through a mechanism that impinges on CIN activity, and thus acts upstream of previously reported LID-attenuating mechanisms such as muscarinic-dependent LTD of D1R MSNs[23] or possible nicotinic receptor desensitization[22].

These findings concord well with the recent reappraisal of Barbeau's DA and ACh seesaw[64,71] and potentially reconcile the perplexing observation that ablation and inhibition of CIN[17,22] as well as augmentation of cholinergic signaling can attenuate LID[22,23]. It is well understood that DAN degeneration and the resulting hypodopaminergic state induces aberrant CIN function which in turn is a key determinant of LID induction[19,20,72]. However, in contrast to the long held assumption that DAN degeneration results in overall increased cholinergic (ACh) tone[73], it was recently observed that animals which undergo DAN ablation show reduced ACh tone compared to controls[64]. The classical view that ACh is elevated following DAN degeneration can in part be explained by the fact that DAN lesioned animals show a greater reduction in DA as compared to ACh. Thus, DAN degeneration appears to produce a hypercholinergic state relative to simultaneous DA signaling, but a dual hypodopaminergic/

hypocholinergic state compared to the healthy basal ganglia. This net reduction in ACh following DAN loss is in line with the previous observation that Shh$_{DAN}^{-/-}$ animals exhibit an eight-fold reduction in striatal ACh concentration compared to controls[38]. Concordant, here we find that CIN of the DLS in Shh$_{DAN}^{-/-}$ animals show decreased expression of neuronal activity marker p-rpS6, which can be normalized by agonists of the Shh effector Smo or transgenic expression of SmoM2 on CIN. Ablation of CIN, which has been shown to block LID[17], might thus prevent the formation of a LID-facilitating gain of function among CIN when Shh$_{DAN}$ to Smo$_{CIN}$ signaling is absent. Indeed, our results point to a close, inhibitory cross talk between dopaminergic and Shh signaling in CIN (Fig. 8e) and suggest that the interruption of Shh$_{DAN}$ to Smo$_{CIN}$ signaling is not merely permissive for LID but is a critical mechanism by which LID-inducing CIN pathology is set in motion following the loss of DAN. A parsimonious possibility is that DA signaling onto CIN becomes unrestrained when Shh$_{DAN}$ to Smo$_{CIN}$ signaling is disrupted and thus facilitates LID.

Shh/Smo signaling is known to act across multiple time scales via both rapid G-protein-dependent signaling and slower transcriptional regulation of target genes[62,74]. The activation of these diverse signaling pathways suggests that Shh$_{DAN}$ to Smo$_{CIN}$ signaling might impinge on several homeostatic mechanisms which help maintain physiological CIN activity and which in turn become dysregulated following DAN degeneration and the loss of Shh$_{DAN}$. The potential action of these downstream signaling cascades is supported by several of our observations in the current experiments. For example, we observed that acute Smo agonist exposure can reduce pERK and boost p-rpS6 levels in CIN within minutes through a mechanism that requires Smo expression on CIN. The evidence laid out across our paradigms further suggests that Smo signaling may functionally oppose the well established D2R mediated inhibition of CIN[29] given that increasing Smo activity relative to DA signaling in CIN attenuated LID while increasing DA signaling by L-Dopa or DAN stimulation facilitated LID (Fig. 8a–d). Concordant, we found that the dose of Smo agonist needed to counteract LID was dependent on the dose of L-Dopa used to induce LID, further suggesting that these pathways inhibit each other in a graded and interdependent manner in CIN (Fig. 8e).

In the context of long-term transcriptional changes, 6-OHDA lesioning or diphtheria toxin mediated ablation of DAN is known to result in decreased spontaneous activity and excitability of CIN despite a simultaneous reduction of inhibitory D2 receptor (D2R) activity[19,29,64]. The long-term reduction of Shh levels in the striatum that occurs alongside DAN degeneration could in part

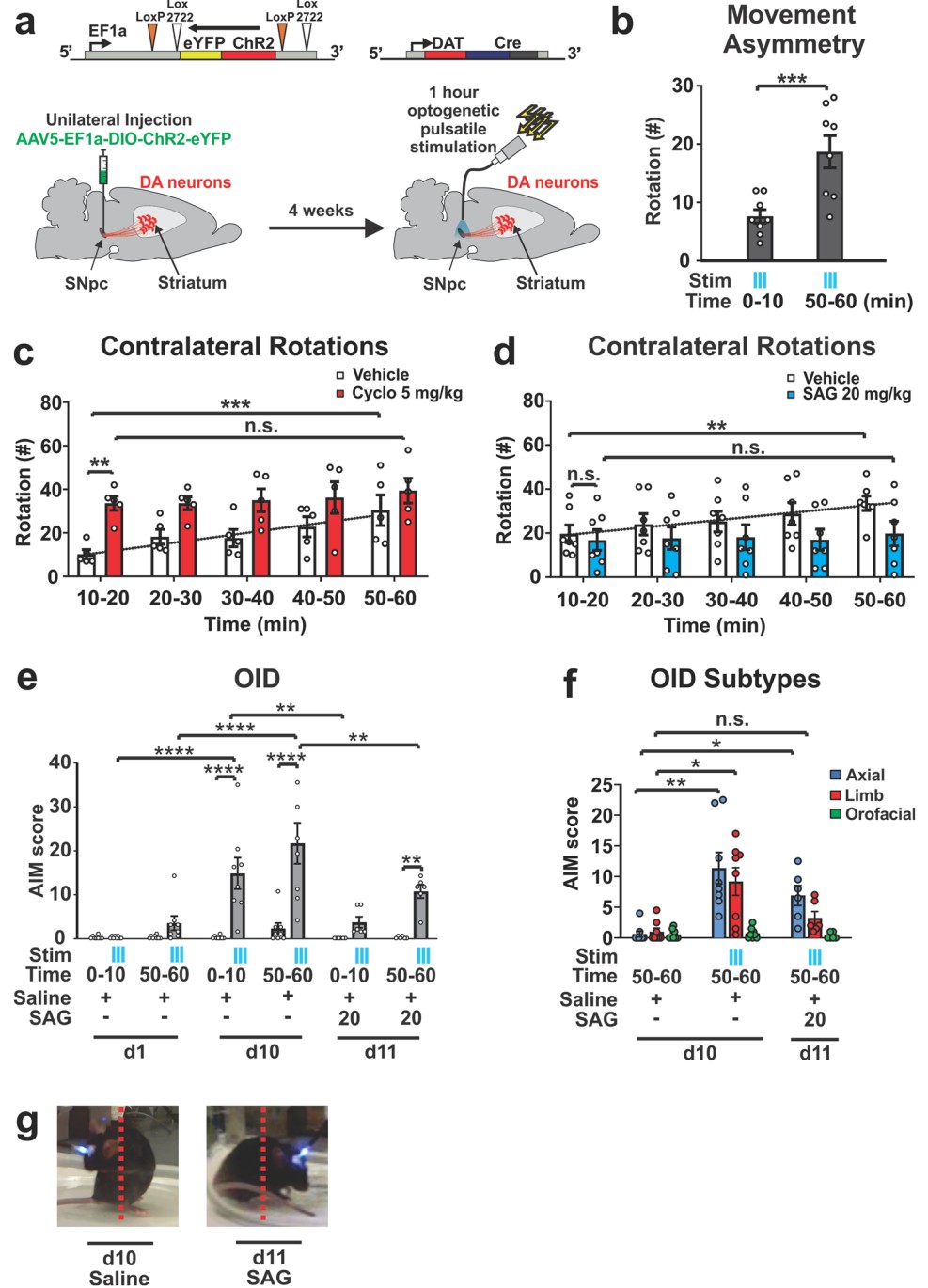

explain this paradoxical finding given that reduced Smo signaling activates transcription of the muscarinic autoreceptor M2[38]. Therefore, changes in transcriptional regulation of muscarinic autoreceptor expression following a loss of $Shh_{DAN}$ to $Smo_{CIN}$ signaling could underlie the long-term depression of tonic CIN activity previously observed in 6-OHDA or diphtheria toxin lesion studies[19,64].

In a recent study, 6-OHDA lesion and subsequent L-Dopa administration produced an increase of tonic CIN firing beyond that observed at normal levels[19]. This observation would suggest that DAN degeneration produces a long-term rearrangement of DA action on CIN and shifts the primary mechanism of DA action from inhibitory D2R to excitatory D5R[29,34]. This activation of CIN by L-Dopa in the hypodopaminergic brain was determined to arise from a reduction in calcium activated

potassium (SK) channel activity which could not be restored by modulators of glutamatergic or GABAergic signaling[19]. The pharmacological normalization of SK channel activity attenuated LID, presumably by restoring CIN properties[19]. Interestingly, SK channels are inhibited by muscarinic M2/M4 autoreceptor signaling[26,75,76] which is increased in the absence of $Shh_{DAN}$ to $Smo_{CIN}$ signaling[38]. Thus, reduced $Smo_{CIN}$ activity following the loss of $Shh_{DAN}$ could depress SK channel activity and cause, in part, the altered electrophysiology observed in CIN following DAN degeneration and L-Dopa administration[19].

Together, our data demonstrates that $Shh_{DAN}$ to $Smo_{CIN}$ signaling attenuates LID in the L-Dopa treated parkinsonian brain and thus provides a strong rationale for exploiting effectors of Smo signaling in CIN for the development of anti-LID medication. Further, it will be of interest to explore whether other side

**Fig. 6 Prolonged optogenetic stimulation of DAN depletes Shh$_{DAN}$ signaling and facilitates LID-like behavior. a** Channelrhodopsin (ChR2) was expressed unilaterally in DAN. Following a 4 week recovery period, DAN were stimulated daily for an hour with 5 s-long bouts separated by 30 s pauses. Bouts of optogenetic stimulation consisted of 10-ms light pulses at 60 Hz. AIMs were scored throughout the hour long session. **b** Stimulation resulted in a contralateral rotational bias which increased across each hour long session ($n = 8$; paired two-tailed Student's $t$ test: ***$p < 0.001$). **c** Number of contralateral rotations in 10 min bins over the course of an hour long stimulation session in animals injected with vehicle (white bars) or cyclopamine (5 mg/kg; red bars; $n = 5$ per condition). RM two-way ANOVA treatment effect: $F_{(1,8)} = 7.40\ p = 0.026$. Post hoc Bonferroni's test: **$p < 0.01$ vehicle vs. cyclopamine (Cyclo); time effect $F_{(4,32)} = 5.69\ p = 0.001$. Post hoc Bonferroni's test: ***$p < 0.001$ veh 10–20 min vs. veh 50–60 min. **d** Contralateral rotations as in **c** following vehicle or SAG (20 mg/kg; $n = 7$ per condition; RM two-way ANOVA, time effect: $F_{(4,44)} = 6.96\ p = 0.0002$; post hoc Bonferroni's test: **$p < 0.01$ veh 10–20 min vs. veh 50–60 min). **e** Optogenetically induced dyskinesia (OID) AIMs were scored in the absence and presence of stimulation (blue ticks) during the first and last 10 min of each stimulation session. Results are reported for days 1, 10, and 11, the last of which, SAG (20 mg/kg) was administered prior to the start of stimulation ($n = 8$; two-way ANOVA, stimulation effect: $F_{(1,38)} = 61.1$, $p < 0.0001$; post hoc Bonferroni's test: ****$p < 0.0001$ stimulation vs. non-stimulation day 10 and **$p < 0.01$ stimulation vs. non-stimulation day 11; time × stimulation effect: $F_{(15,38)} = 8.59$, $p < 0.0001$; post hoc Bonferroni's test: ****$p < 0.0001$ stimulation during day 1 vs. day 10 and **$p < 0.01$ stimulation during day 10 vs. day 11). **f** Quantification of axial (blue), limb (red), and orofacial (green) subcategories of OID-AIMs. Axial AIMs were unresponsive to SAG treatment while limb AIMs were reduced to unstimulated control levels in response to SAG ($n = 6$–8; two-way ANOVA stimulation/treatment effect: $F_{(2,19)} = 10.41$, $p = 0.0009$. AIM subtype effect: $F_{(1.875,35.63)} = 19.55$, $p < 0.0001$. Stimulation/treatment × AIM subtype effect: $F_{(4,38)} = 6.78$, $p = 0.0003$. Tukey's multiple comparisons test: **$p < 0.001$ axial AIM for no stimulation day 10 vs. stimulation day 10, *$p < 0.05$ axial AIM for no stimulation day 10 vs. stimulation + SAG day 11, limb AIM for no stimulation day 10 vs. stimulation + SAG day 11, n.s. $p = 0.207$ limb AIM for no stimulation day 10 vs. stimulation + SAG day 11. **g** Still video images of mouse posture during stimulation with and without SAG (20 mg/kg) injection All graphs are plotted as average ± SEM.

effects of L-Dopa treatment, like impulse control diseases, or L-Dopa treatment refractory Parkinson's disease signs like cognitive deficits[77,78] can be attenuated by manipulating the balance between Shh$_{DAN}$/Smo$_{CIN}$- and DA signaling. Finally, the continued exploration of how Shh$_{DAN}$ influences physiological CIN processing, particularly through acute temporal fluctuations in Smo signaling, might be of interest for studies of basal ganglia dependent learning and motor coordination.

## Methods

**Mice**. All mice were maintained on a 12 h light/dark cycle, with ad libitum food and water. Animal use and procedures were in accordance with the National Institutes of Health guidelines and approved by the Institutional Animal Care and Use Committees of the City University of New York. All experiments were carried out in young adult animals beginning at 2 months of age and weighing between 22 and 28 g unless stated otherwise. Males and females from a mixed CD1 and C57/BL6 background were used for all experiments in approximately equal proportions.

C57/Bl6 mice, JAX # 000664, were purchased from Jackson Laboratory.

$AK^{-/-}$ (Pitx3$^{ak/ak}$) mice[55,56] and were generously provided by Dr. Un Jung Kang (NYU), propagated by homozygous breeding and genotyped using primers in Supplementary Table 3.

Shh$_{DAN}$$^{-/-}$ mice[38] and controls were produced by crossing Shh-nLacZ$^{L/L}$ females with Shh-nLacZ$^{L/+}$/Dat-Cre males resulting in Shh-nLacZ$^{L/L}$/Dat-Cre experimental (Shh$_{DAN}$$^{-/-}$) and Shh-nLacZ$^{L/+}$/Dat-Cre (Shh$_{DAN}$$^{-/+}$) control mice and genotyped using generic Cre and Shh primers in Supplementary Table 3.

Smo$_{CIN}$$^{-/-}$ mice[79] and controls were produced by crossing Smo$^{L/L}$ females with Smo$^{L/+}$/ChAT-IRES-Cre males[80] resulting in Smo$^{L/L}$/ChAT-IRES-Cre (Smo$_{CIN}$$^{-/-}$) experimental and Smo$^{L/+}$/ChAT-IRES-Cre (Smo$_{CIN}$$^{-/+}$) control mice. Smo $^{L/+}$(JAX # 004526) and ChAT-IRES-Cre (JAX # 006410) mice were purchased from Jackson Laboratory and genotyped using the Chat-Cre, Smo$^{fl/fl}$, and Smo WT primers in Supplementary Table 3.

SmoM2$^{+/-}$$_{CIN}$ $AK^{-/-}$ ($^L$STOP$^L$SmoM2-YFP/ChAT-IRES-Cre /Pitx3$^{ak/ak}$) mice and are maintained by crossing Pitx3$^{ak/ak}$ females with triple heterozygous $^L$STOP$^L$SmoM2-YFP[81]/ChAT-IRES-Cre /Pitx3$^{ak/+}$ males resulting in $^L$STOP$^L$SmoM2-YFP/ChAT-IRES-Cre /Pitx3$^{ak/ak}$ (SmoM2$^{+/-}$$_{CIN}$ $AK^{-/-}$) experimental—and ChAT-IRES-Cre /Pitx3$^{ak/ak}$ ($_{ChAT-Cre}$ $AK^{-/-}$)—control mice. Tissue specific activation of SmoM2 was confirmed by eYFP expression. $^L$STOP$^L$SmoM2-YFP mice (JAX # 005130) were purchased from Jackson Laboratory and genotyped using the ChAT-Cre, SmoM2 mutant, and SmoM2 WT primers in Supplementary Table 3.

**Macaques**. The non-human primate experiments were performed in accordance with the European Union directive of September 22, 2010 (2010/63/EU) on the protection of animals used for scientific purposes in an AAALAC-accredited facility following acceptance of study design by the Institute of Lab Animal Science (Chinese Academy of Science, Beijing, China). The four Macaca fascicularis male monkeys (Xierxin, Beijing, PR of China) were housed in individual cages allowing visual contact and interactions with other monkeys in adjacent cages. Food and water were available ad libitum. Animal care was supervised daily by veterinarians skilled in the healthcare and maintenance of NHPs.

**Antibodies**. Anti-tyrosine hydroxylase (TH) (1:500; RRID: AB_657012) and anti-ChAT (1:100; RRID: AB_144P) were purchased from Milipore. Anti-phospho-Erk1/2 (1:400; RRID: AB_9101), and anti-phospho-rpS6 (ser240/244; RRID: AB_5364) were from Cell Signaling Technology. Anti-NeuN (1:200; RRID: AB_MAB377) from Chemicon International. Various Alexa Fluor antibodies were used at 1:250 dilutions for immunohistochemistry (Jackson Immuno Research).

**Drugs**. In mice, all pharmacological agents were administered via intraperitoneal (i.p.) injection. Animals were treated daily in a volume of 10 ml/kg of body weight with a combination of L-Dopa (5–25 mg/kg; Sigma-Aldrich D1507) and the peripheral L-amino acid decarboxylase antagonist, benserazide (12.5–20 mg/kg; Sigma-Aldrich B7283) diluted in 0.9% sterile saline (referred to as just L-Dopa and specified doses described in main text). Sonic Hedgehog agonist (SAG Carbosynth Limited FS27779; 0.8–20 mg/kg) and antagonist (cyclopamine Carbosynth Limited FC20718; 2.5–5 mg/kg) were dissolved in DMSO and brought into solution with 45% HPCD (Sigma-Aldrich H107) in 0.9% sterile saline for 6-OHDA and $AK^{-/-}$ AIM studies. SAG-HCl (Carbosynth Limited FS76762) was diluted in 0.9% sterile saline and used for the Shh$_{DAN}$$^{-/-}$ and Smo$_{CIN}$$^{-/-}$ mouse studies. Both cyclopamine and SAG were administered 15 min before L-Dopa injections. Amantadine hydrochloride (Sigma-Aldrich A1260) was dissolved in 0.9% sterile saline and given 100 min prior to L-Dopa administration. M4-PAM (VU0467154 StressMarq Biosciences SIH184) was dissolved in 0.9% sterile saline and given at the same time as L-Dopa. pPurmorphamine (ABCam AB120933) was dissolved in a cocktail of Polyethylene glycol and ethanol in PBS and given 15 min before L-Dopa dosing. Control mice were treated with carrier (DMSO, HPCD, etc.) in similar proportions to the treatment drugs.

**Mouse unilateral 6-OHDA model[82]**. Mice were anesthetized with a mixture of ketamine (80 mg/kg) and xylazine (12 mg/kg) administered via i.p. and the surgical field was prepared with betadine. Bupivacaine (Marcaine Sigma-Aldrich B5274), a local anesthetic, was subcutaneously (s.c.) injected near the incision site. Bregma was visualized with 3% hydrogen peroxide. Two unilateral injections (2 × 2 μl each) of 6-OHDA were administered into the left striatum using the following coordinates based on the Mouse Brain Atlas by Paxinos and Franklin (2001): Anteroposterior (AP) + 1.0 mm; Lateral (ML), +2.1 mm; Dorsoventral (DV), −2.9 mm; and AP + 0.3 mm; L + 2.4 mm; DV −2.9 mm. 6-OHDA-HCl (Sigma-Aldrich H4381; 3.0 mg/ml) was dissolved in a solution containing 0.2 g/l ascorbic acid (Sigma-Aldrich A92902) and 9 g/l NaCl and injected via a Hamilton syringe with a 33-gauge needle attached to a micro-syringe pump (World Precision Instruments) at a rate of 0.4 μl/min and left in place for 3 min afterwards before being retracted slowly. Following surgery, animals were injected with 5% sucrose (10 ml/kg, s.c.) and saline (10 ml/kg, i.p.) and recovered on a heating pad. To avoid dehydration and weight loss, hydrogel pouches and high-fat chow were given to the mice ad lib. Behavioral testing and drug treatment began 3 weeks following surgery. Lesion assessments were quantified by ipsilateral paw-use bias in the cylinder test and verified histologically, postmortem, by quantification of TH fiber density at the end of experiments. Only animals with 70% or more of TH positive fiber depletion were included in the analyses.

**Forelimb cylinder test**. The cylinder test was used to assess the anti-akinetic effects of L-Dopa in 6-OHDA lesioned mice by measuring forelimb paw placement. Mice were placed in a glass cylinder (10 cm wide × 14 cm high) 30 min post

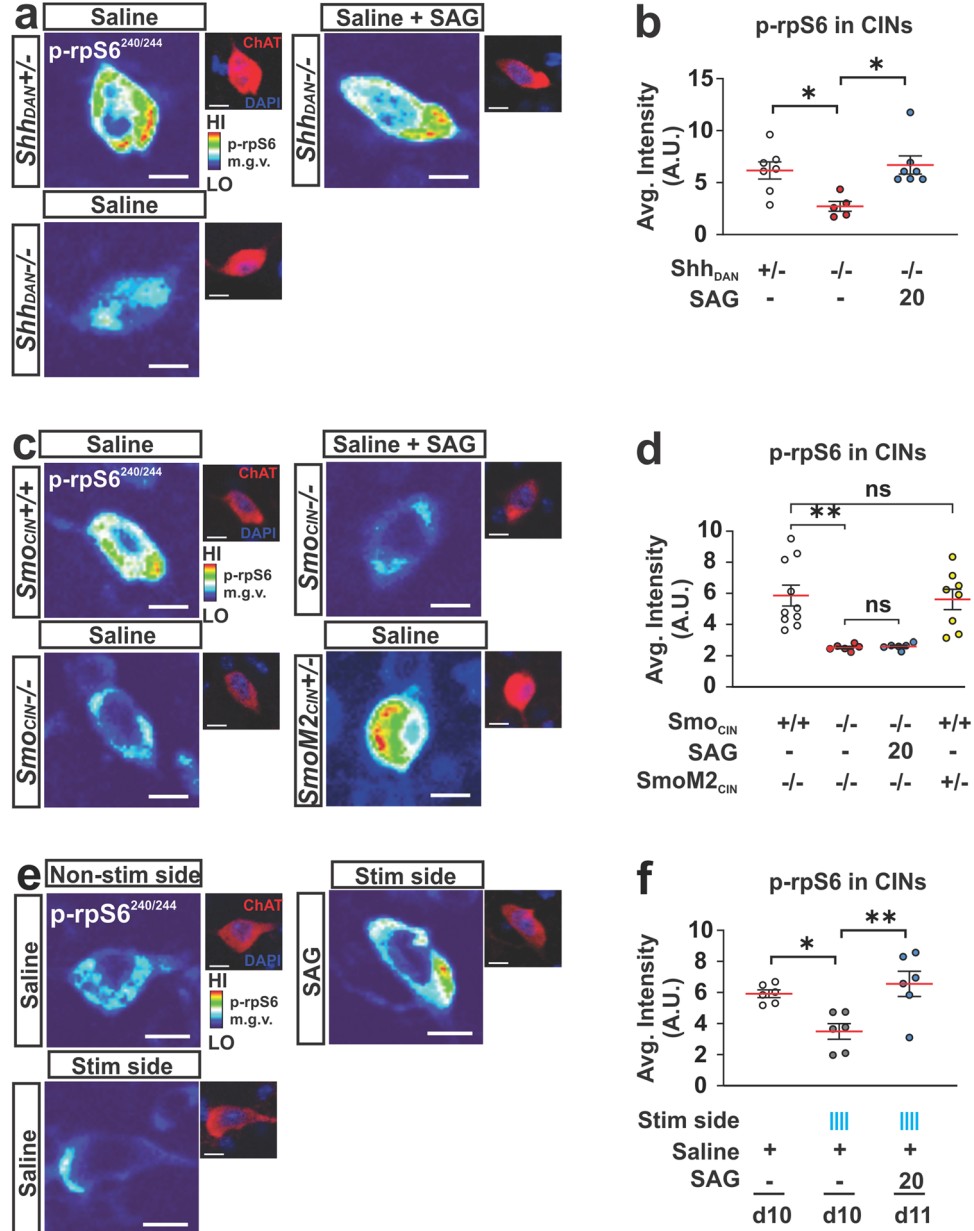

**Fig. 7 Shh$_{DAN}$ signaling impinges on the activity of CIN via Smo$_{CIN}$. a** Representative heat maps showing DLS CIN levels of the ribosomal activity marker p-rpS6$^{240/244}$ in Shh$_{DAN}$$^{-/-}$ mice, Shh$_{DAN}$$^{+/-}$ littermate controls, and in response to a single dose of SAG (20 mg/kg; p-rpS6$^{240/244}$ levels were normalized to NeuN levels in CIN identified by ChAT staining, see smaller images; scale bar = 10 μm). **b** Quantification showed reduced average intensity of CIN p-rpS6$^{240/244}$ staining in Shh$_{DAN}$$^{-/-}$ mice compared to Shh$_{DAN}$$^{+/-}$ littermate controls which was rescued by a single dose of SAG (20 mg/kg; n = 5–7 per genotype and drug condition, 60–100 cells per n; each dot represents one biological replicate; red lines indicate the mean and black lines indicate SEM; one-way ANOVA followed by Tukey's multiple comparisons test: *p < 0.05 controls vs. mutants, or mutants with and without treatment). **c** Representative heat maps showing DLS CIN levels of p-rpS6$^{240/244}$ in Smo$_{CIN}$$^{-/-}$ mice (loss of function), SmoM2$_{CIN}$$^{+/-}$ mice (gain of function), Smo$^{+/+}$$_{ChAT-cre}$$^{+/-}$ littermate controls, and in Smo$_{CIN}$$^{-/-}$ mice administered SAG (20 mg/kg; scale bar = 10 μm). **d** Quantification showed reduced average CIN levels of p-rpS6$^{240/244}$ in Smo$_{CIN}$$^{-/-}$ mice compared to Smo$^{+/+}$$_{ChAT-cre}$$^{+/-}$ littermate controls which could not be rescued by SAG treatment (n = 6–10 per genotype and treatment group, 60–100 cells per n; one-way ANOVA followed by Tukey's multiple comparisons test: **p < 0.01 controls vs. Smo$_{CIN}$$^{-/-}$, or SmoM2$_{CIN}$$^{+/-}$ vs. Smo$^{-/-}$$_{ChAT-cre}$$^{+/-}$. n.s. indicates p > 0.05). **e** Representative heat maps showing DLS CIN levels of p-rpS6$^{240/244}$ in response to optogenetic stimulation of DAN (scale bar = 10 μm). **f** Quantification showed reduced average levels of p-rpS6$^{240/244}$ in CIN of the DLS after 1 h of pulsatile DAN stimulation compared to non stimulated controls also expressing ChR2 in DAN. This reduction was elevated to control levels by a single dose of SAG (20 mg/kg) injected prior to optogenetic stimulation (n = 6 per condition; 40–100 cells per n; one-way ANOVA followed by Tukey's multiple comparisons test: *p < 0.05, **p < 0.01).

L-Dopa injection and then independent forelimb contacts with the cylinder were counted for 4 min. Two mirrors were placed in the back to observe mice from all angles. The limb use asymmetry score was expressed as a ratio of contralateral to the lesion paw use over the total number of wall contacts with both paws[83].

**Open field test.** In the open field test, whole body movements were measured for 5 min using a Noldus Ethovision XT video tracking system. The animals head, body, and tail positions were tracked in a transparent Plexiglas 4-chamber box apparatus (50 × 50 cm for each chamber). Spontaneous locomotion was quantified as distanced traveled at baseline (no drug) and

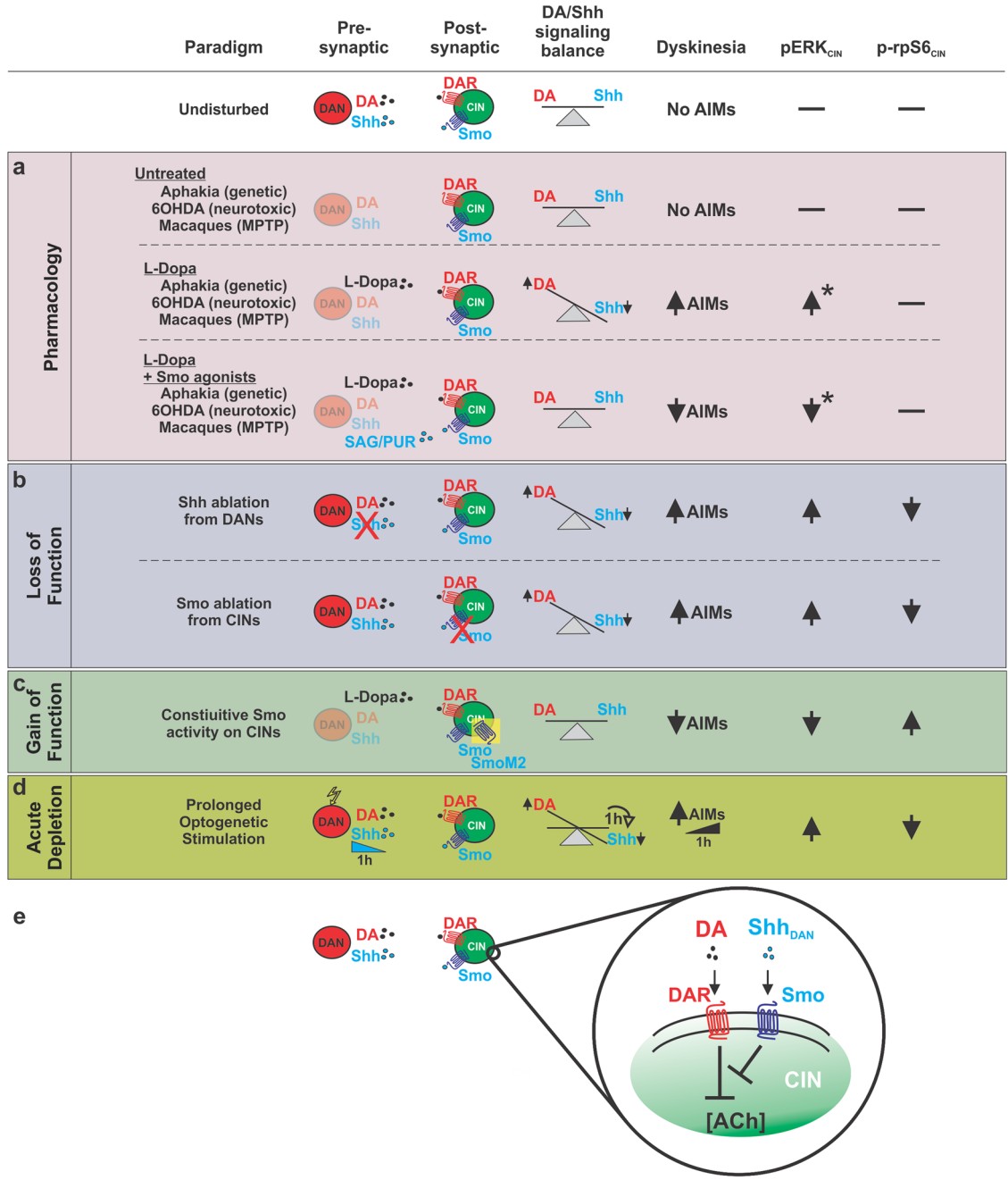

**Fig. 8 Manipulation of Shh signaling between DAN and CIN modulates LID, pERK, and p-rpS6 across multiple paradigms spanning time scales and degrees of DAN denervation. a–d** Across the animal models of Parkinson's Disease and experimental paradigms utilized in this study, a relative imbalance between Shh (low) and DA (high) signaling onto CIN was consistently observed to induce LID-like pathology. This pathology was accompanied by decreases in neuronal activity marker p-rpS6 and increases in LID histological marker pERK among CIN of the DLS. States defined by relatively low Shh and high DA signaling onto CIN facilitated LID. Postsynaptic manipulations of Smo on CIN produced effects mimicking the complementary pharmacological and presynaptic manipulations suggesting CIN are the relevant cell type through which Shh-dependent modulation of LID occurs. DAN are the physiological source of Shh on which CIN depend and ablation of Shh from DAN produced LID-like behavior and changes in CIN pERK and p-rpS6 levels. Further, not only was acute Smo pharmacology sufficient to rapidly attenuate LID following SAG administration, but prolonged optogenetic stimulation of DAN led to a diminishment of Shh signaling within the course of a single hour and produced behavior resembling that observed in LID. Thus the relative balance between DA and Shh signaling onto CIN is not only perturbed following drastic denervation of DAN, but can vary at shorter time scales. **e** The findings derived from these distinct paradigms indicates that the Shh/Smo signaling pathway counteracts DA signaling in CIN. *Changes in pErk determined from examination of murine models.

45 min following drug administration. Asymmetry of locomotion was calculated as a percentage of lesion ipsilateral to contralateral turning bias.

**Abnormal involuntary movements (AIMS).** For the 6-OHDA model we quantified four types of AIMs: (1) abnormal rotational locomotion, (2) axial AIMs,

showing dystonic posturing and severe twisting of the head or neck, (3) limb movements with rapid, jerky movements of the front or hind limbs, and (4) orofacial movements of abnormal chewing, licking, grooming, or sticking out of the tongue. The four types of AIMs were scored on a severity scale of 0–4, with 0 exhibiting no abnormality and 4 showing uninterruptable dyskinetic movement.

1–2 blinded experimenters rated AIMs. AIM scores were assessed 35 min post treatment injection and 20 min post L-Dopa injection. Animals were single caged during AIM assessments. Scores for each animal were recorded every 20 min, for 1 min each, over 2 h. The total sum of all four AIM types scores throughout 2 h was calculated as the Total AIM score.

For the $AK^{-/-}$ mouse model we acquired AIMs scores differently compared to 6-OHDA lesion model because $AK^{-/-}$ mice have a bilateral reduction in dopaminergic innervation of the striatum and thus do not exhibit unilateral turning behavior. Instead, $AK^{-/-}$ mice were recorded in a clear plastic cylinder (10 cm in diameter and 12.5 cm in height) 30 min following either saline (baseline day) or L-Dopa (test day) injection. Mirrors were placed behind the cylinder to allow for ventral views of the behavior. Each trial was 5 min with a 30-s habituation period followed by behavioral scoring. The "total AIM duration" was quantified for each trial as the length of time the animal displayed sliding or shaking of the forelimb paws while rearing on the cylinder wall. "Three-paw" dyskinesias were also recognized as events with shaking of two forelimbs and one hindlimb. "Four-paw" dyskinesias were observed in very extreme dyskinetic cases when the animal balanced on its tail for brief moments.

**Primate MPTP model.** The 1-methyl-4-phenyl-1,2,3,6-tetrahydropyridine (MPTP) intoxication protocol, chronic L-Dopa treatment, and the clinical assessments were conducted in four male macaques (Macaca fascicularis, Xierxin, Beijing, PR of China). The macaques were first rendered parkinsonian with MPTP-hydrochloride (0.2 mg/kg, i.v., Sigma) dissolved in saline. Daily (at 9 a.m.) assessment of parkinsonism in home cages for 30 min by two blinded observers was done using a validated rating scale assessing tremor, general level of activity, body posture (flexion of spine), vocalization, freezing and frequency of arm movements (for each upper limb), and rigidity. Once parkinsonism was stable, levodopa (Madopar, Roche, Levodopa/carbidopa, ratio 4:1) was administered twice daily for 4–5 months at an individually tailored dose (18–22 mg/kg) designed to produce a full reversal of the parkinsonian condition (p.o. by gavage). Over this period, animals developed severe and reproducible dyskinesia, presenting choreic–athetoid (characterized by constant writhing and jerking motions), dystonic, and sometimes ballistic movements (large-amplitude flinging, flailing movements) as seen in long-term L-Dopa-treated Parkinson's disease patients. SAG (3, 9, 27 mg/kg) was dissolved in 10% DMSO, 45% HPCD in 0.9% saline and administered i.v. Within-subject escalation was performed with a washout period of 3 days between escalating doses.

Immediately after drug administration, monkeys were transferred to an observation cage (dimensions—1.1 m × 1.5 m × 1.1 m) as per guidelines[84]. The total duration of observation was 240 min of drug exposure. We performed a battery of behavioral observations as previously described[85]. Experts blinded to the treatment observed 10-min video recordings taken every 30 min throughout the duration of the experiment and scored the severity of the parkinsonian condition using the parkinsonian disability score. The parkinsonian disability score is a combination of four different scores: (1) the range of movement score, (2) bradykinesia score, (3) posture score, and (4) tremor score. These four scores are combined using formula: $(4 - \text{range of movement}) + \text{bradykinesia} + \text{postural abnormality} + \text{tremor}$. We rated the severity of dyskinesia using the Dyskinesia Disability Scale[84]: 0, dyskinesia absent; 1, mild, fleeting, and rare dyskinetic postures and movements; 2, moderate, more prominent abnormal movements, but not interfering with normal behavior; 3, marked, frequent and, at times, continuous dyskinesia intruding on the normal repertoire of activity; or, 4, severe, virtually continuous dyskinetic activity replacing normal behavior and disabling to the animal.

Primate behavior was rated using most recent guidelines[84] which include the recommendation to use nonparametric tests (nature of the rating scale and number of individuals). Nonparametric analysis is less sensitive compared to parametric analysis and reduces the risk of identifying false-positive responses. We presented the time course of parkinsonian disability and dyskinesia scores in 30 min time bins over the 4 h observation period. We also presented the median of the total scores of disability, dyskinesia, chorea, and dystonia at 0–2 h following treatment. We statistically compared the parkinsonian and dyskinesia scores between different conditions using a Friedman's test followed by Dunn's multiple comparison test.

**Optogenetic manipulations.** Implants and patch cables were constructed and polished[86]. Mice were anesthetized with isoflurane and the surgical field was prepared with betadine. Bupivacaine was injected in the scalp prior to incision. Bregma was visualized with 3% hydrogen peroxide and a craniotomy was made over the injection site. The AAV5-EF1a-DIO-hChR2(H134R)-eYFP-WPRE virus (UNC Vector Core) was injected at AP −3.2 mm, ML +1.5 mm, DV −4.3 mm by pressure injection with a pulled glass pipette and allowed to dwell for 10 min. A ferrule implant was placed at AP −3.2 mm, ML +1.5 mm, DV −4.2 mm, secured with metabond, and protected with a dust cap (Thor Labs). Following surgery, mice were given 0.05 mg/kg buprenex and recovered on a heating pad. At least 1 month was allowed for viral expression.

One trial of stimulation consisted of a 10-min habituation period followed by an hour of intermittent stimulation for 5 s in which 10 ms long pulses at 60 Hz and a laser power of 20 mW as measured at the tip of the implant were produced, followed by a 30 s pause. Stimulation was controlled via a TTL pulse from Ethovision's software to a Doric four channel pulse generator controlling a 710 nm

Laserglow DPPS laser. All sessions were recorded and analyzed with Ethovision. Pharmacological agents were given 30 min prior to the beginning of the trial.

For optogenetic-induced dyskinesia (OID) experiments, stimulation occurred in a 5 l glass cylinder and consisted of 1 h of intermittent stimulation described in optogenetic stimulation paradigms section. Stimulation occurred daily for the duration of the experiments. A second camera (Casio EX-FH100) recorded behavior from the side of the cylinder during the first and last 10 min of the trial. AIMs were assessed on limb, orofacial, and axial movements using the same scoring scale developed for the unilateral 6-OHDA lesion paradigm[87]. AIMs were scored separately during stimulation and non-stimulation periods. Pharmacological agents were given 30 min prior to the beginning of the trial.

**Immunohistochemistry.** Animals were anesthetized deeply with pentobarbital (10 mg/ml) 30 min after last drug injection and transcardially perfused with 50 ml of 4% paraformaldehyde (PFA) in 0.2 M phosphate buffer pH 7.2. The brains were extracted and fixed overnight in 4% PFA and equilibrated in 30% sterile sucrose for 48 h. Brains were mounted in OTC and cryo-sections produced at 40 μm. Sections were washed in PBS for 10 min, permeabilized in PBS with 0.3% triton (PBST) for 10 min, blocked in PBST with 10% horse serum (blocking buffer) for 1 h, incubated overnight at 4 °C in primary antibody diluted in blocking buffer, washed for 10 min in PBS 3x, intubated in secondary antibody diluted in blocking buffer for 1.5 h, washed for 10 min in PBS 3x, and then washed with a 1:5000 DAPI solution for 5 min[88]. Sections were mounted in Vectashield (Vector Laboratories Inc.). All steps were at room temperature unless noted.

Confocal microscopy was performed using a Zeiss LMS 810 laser-scanning confocal microscope using the same acquisition parameters for experimental and control tissue for each experiment.

TH immunofluorescence intensity was quantified in the striatum with MacBiophotonics ImageJ, and the data represented mean gray levels above background.

The fraction of pErk$^{1/2}$ positive CINs was expressed as a percentage of pErk$^{1/2}$ positive CIN out of the total number of CIN in each of 375 × 375 μm region of interest (ROIs) in the dorsolateral and DMS of each coronal section analyzed. Relative position of the ROI box was kept the same between genotypes and treatment groups and along the anterior posterior axis. NeuN fluorescence intensity of the same cell was used for normalization of fluorescence across treatment and genotype groups. Quantification was performed blinded to the treatment and/or genotype of the subject in three striatal slices: anterior (AP: ~1.34 mm), middle (AP: ~0.74 mm), and posterior (AP: ~0.14 mm) per animal. Co-stained and single stained cells were manually counted using FIJI/ImageJ software (NIH).

Each data point reflects the average percent of pErk$^{1/2}$ in one animal.

**p-rpS6$^{240/244}$ analysis.** The mean gray value of p-rpS6$^{240/244}$ intensity was measured in FIJI/ImageJ software (NIH) using a freehand ROI area delineating the soma of each ChAT stained CIN in a 375 × 375 mm box in the DLS[70]. NeuN fluorescence intensity of the same cell was used for normalization of fluorescence across treatment and genotype groups. Heat map images reveal the intensity levels of p-rpS6$^{240/244}$ fluorescence based on a 16 pseudocolor palette. In total, 200–300 CINs were quantified per each condition. Each data point reflects the average level of p-rpS6 intensity in one animal.

**Statistics and reproducibility.** No statistical methods were used to predetermine sample size.

Multiple comparisons were analyzed using two-way, three-way, or repeated measures ANOVA with GraphPad Prism 8.0 software, followed by post hoc Bonferroni's test for specific comparisons. For two-group comparisons, two-tailed paired or unpaired Student's t test was used. Nonparametric statistics were used to verify all main effects using Dunnett's multiple comparison test or Mann–Whitney test when appropriate.

**Reporting summary.** Further information on research design is available in the Nature Research Reporting Summary linked to this article.

## Data availability

The data that support the findings of this study are available within the article and its Supplementary Information files and Supplementary Data 1, or are available from the corresponding author upon reasonable request.

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

## Acknowledgements

This work was supported by the American Parkinson Disease Association, Research Foundation of the City University of New York, and NIH NS095253 to A.H.K., and in part by NIH R25GM56833 (P.I. Mark Steinberg) via L.M.'s "RISE" fellowship and NIH T32GM136499 (P.Is. Ruth Stark and Mark Steinberg) via S.U.-C.'s "G-RISE" fellowship. L.M., D.R.Z., S.U.-C., and L.S. acknowledge the administrative, travel, and mentoring support they received through the biology graduate programs of the Graduate Center of the City University of New York and the CUNY School of Medicine. A.H.K. thanks John Martin for active mentoring and support. We are grateful to Y.J.Z., L.H., C.Y.L., and X.R.L. for the care of the non-human primates. We thank Jeff Beeler, Paul Forlano, Christoph Kellendonk, John Martin, Stephen Rayport, and Luis Reyes-Gonzalez for their critical comments on earlier versions of the manuscript.

## Author contributions

L.M., D.R.Z., and A.H.K. designed the mouse experiments. L.M. performed all LID related behavioral and cytohistochemical studies in mouse with contributions from S.U.-C., H.R., L.S., and E.F. in data analysis. D.R.Z. developed the optogenetic Shh exhaustion paradigm. C.Q., Q.L., and E.B. conducted the NHP experiments and analyzed the associated data. S.U.-C., performed cytohistochemical analysis of DAN integrity across mouse lines. L.M., D.R.Z., S.U.-C., and A.H.K. wrote the manuscript with E.B. contributing the description of the NHP study. A.H.K. coordinated and supervised all aspects of the project.

## Competing interests

The authors declare no competing interests.
