## [Peer Review File · Communications Biology]

Reviewers' comments:

Reviewer #1 (Remarks to the Author):

Cholinergic interneurons (CINs) in the dorsal striatum are implicated in L-Dopa induced dyskinesia (LID), however the precise mechanisms underlying LID are unclear. In this manuscript, Malave and colleagues show that LID can be attenuated by increasing smoothed signaling in 6-OHDA or aphakia KO (AK^{-/-}) mice. They also report a modest decrease in Dyskinesia Score in MPTP-exposed macaques following treatment with 27 mg/kg of SAG. In LID model mice, inhibition of smoothed signaling with cyclopamine increased the number of CINs with positive pERK1/2 staining, whereas SAG treatment had the inverse effect. The authors show that conditional genetic ablation of Smo from CINs exacerbates abnormal involuntary movements (AIMs) in L-dopa treated mice. By contrast, constitutive activation of Smo signaling, specifically in CINs, prevents AIMs in AK^{-/-} mice. Upstream of these events, Malave and colleagues show ablation of Shh from dopaminergic neurons increases AIMs in mice. Similarly, depleting Shh signaling from dopaminergic neurons through prolonged stimulation increases AIMs. The authors also report that Shh Signaling from dopaminergic neurons is necessary for proper CIN physiology, as determined by loss of phospho-rpS6 intensity in CINs upon dopaminergic Shh ablation. Altogether, their study convincingly shows that the onset of AIMs in mice can be attenuated by increasing sonic hedgehog signaling in CINs. Publication of this interesting and important manuscript is strongly supported after the following points have been addressed:

- 1) Please explain the rationale for reporting total AIM score in Figure 1B versus AIM duration in Figure 1C. Is duration included in the methods?
- 2) The decrease in dyskinesia score in macaques following SAG treatment is not as convincing as the results shown for mice—why was a different statistical test used and please discuss why response may have been different
- 3) Cholinergic interneuron pERK1/2 staining is scored as either positive or negative in Figure 1H-I and Extended Figure 5. If possible, the authors should report normalized pERK1/2 fluorescence intensity in CINs, analogous to their measurement of phospho-pS6 intensity in Figure 5.
- 4) Data points in Figure 1H-I, Extended Figure 5, and Figure 5 B,D,F, are displayed as total cells instead of animals per genotype. Our statistical experts tell us that they should be reported by animal and displayed by animal rather than total cells counted.
- 5) If possible, the authors should determine if LID attenuation by Smo signaling occurs via the canonical Hedgehog pathway or by another pathway. For example, is Gli1 or another classical pathway marker changed in CINs following SAG treatment, decreased in CINs by cyclopamine, or relatively decreased in AK^{-/-} mice when compared to non-transgenic littermates?

Minor points:

- 1) Define M4PAM in the main text.
- 2) Use consistent abbreviation for cholinergic (ACh) throughout the text.
- 3) Extended data Fig2C—is a linear R² response expected when the x axis is nonlinear?

Reviewer #2 (Remarks to the Author):

Malave and colleagues present a novel, substantial evidence for the role of cholinergic interneurons of the striatum in the pathophysiology of levodopa-induced dyskinesia (LIDs), a disabling complication of long-term dopaminergic therapy commonly observed in parkinsonian patients. This elegant, multidisciplinary work comes from an excellent group of scientists, experts in the field, which makes this paper relevant and timely, and represents a fundamental step for clarifying the pathophysiology of Parkinson's disease. By shedding new light on the role of SSh signalling pathways, a previously underestimated mechanism, this work has a potential translational relevance and could open the way to novel pharmacological treatments. The authors provide an accurate description of the role of Sonic Hedgehog (Ssh) co-signaling with

dopamine in dopaminergic target areas, specifically the striatum. They show that reduced Shh signaling in dopamine-depleted animals induces expression of LID, whereas restoring the balance of dopamine and Shh signaling by agonists of the Shh signaling effector receptor (Smo) reduced dyskinesias caused by repeated levodopa treatment both in mouse and macaque models of Parkinson's disease. Yet, expression of constitutively active Smo (SmoM2) in cholinergic interneurons induces a resistance to develop LIDs in the aphakia model of PD. By utilizing three well-established PD animal models, the authors developed novel genetic and optical stimulation paradigms for manipulating the strength of dopamine and Shh co-transmission by dopaminergic cells onto striatal cholinergic interneurons in the manifestation of dyskinesias. In both animal species, they observed that dyskinesias are caused by a reduction of Shh-dopamine/Smo signaling on cholinergic targets, and conversely, that enhancing Smo signaling in these interneurons attenuates the involuntary movements through a mechanism that might involve a direct modulation of the striatal output pathways. Phosphorylation of Erk in the striatum is considered a LID marker. Modulation with Smo agonists in cholinergic interneurons was able to modify the occurrence of LIDs. Of note, neither the chronic nor the acute dosing with SSH agents (SAG) affected the prokinetic efficacy of levodopa.

There are some points that require clarification, that would strengthen the significance of these findings.

Major points:

- Clinical evidence does not support the efficacy of anticholinergic drugs in the management of dyskinesia. A relevant question is whether anticholinergics worsen dyskinesias. The authors use M4 PAM which is hypothesized to work on MSN (Fig 7). However, M4 autoreceptor activation on cholinergic interneurons is a well established mechanism of autoinhibition and expected to cause a decrease in ACh release, and thus of cholinergic tone. This should be taken into consideration.
- The role of nicotinic receptors that are known to promote DA release should be included in the proposed scenario (Threlfell et al 2012, Neuron; Brimblecombe et al., 2018 eNeuro).
- Is there a specific effect for striatal targets? Does any other DAN target region express SSH-coupled receptors? Did the authors observe such specificity?
- The authors claim that ERK changes are observed in cholinergic cells. However there is abundant literature showing that spiny neurons are preferentially affected, in particular those expressing D1 receptors (Cholinergic interneurons express D5 receptors). Indeed, spiny neurons of the direct pathway display a dramatic increase in cyclic AMP-responsive element binding (CREB), AP-1, and ERK-dependent gene expression (Heiman et al., 2014 PNAS).
- Prolonged optogenetic activation is expected to potentially deplete other neurotransmitters as well. Did the authors rule out this possibility?

Minor points.

- some typos throughout the text, please amend.
- "it appears the unphysiological CIN activity which emerges in LID following dopaminergic denervation bestows a LID-inductive function onto CINs which can be disengaged through their ablation." Please rephrase to simplify this sentence.

Reviewer #3 (Remarks to the Author):

In this study, Malave and colleagues examine the impact of sonic hedgehog (shh) signaling in the dorsal striatum (with distinguishing dorsolateral and dorsomedial striatum) in L-Dopa induced dyskinesia (LID) animal models. The authors show that agonists (SAG and PUR) of the shh effector Smoothed (Smo) decrease the severity of LIDs. In line with this, they show (with elegant transgenic approaches) that decreasing and increasing ssh signaling (acting on the source, the dopaminergic neurons, or the target, the striatal cholinergic interneurons) allow to increase and decrease, respectively, the severity of LID.

Overall, this constitutes an impressive amount of work, providing in-depth analysis of a crucial problem for parkinsonian patients: how to alleviate the LID (besides DBS which is a successful symptomatic therapy but a very invasive procedure). The data are very interesting and novel. Despite a huge body of experiments (notably the impressive use of sophisticated transgenic mouse lines), some questions remain open and some experiments do not fully support the very strong

conclusion brought by the authors about the effect of Shh signaling for alleviating LID.

Major comments:

1) The authors used in this study a murine model of Parkinson's disease: the AK^{-/-} mice. In Figure 1 they also used for specific subtypes of experiments another rodent model, the 6-OHDA-lesioned mice (for testing SAG and PUR and examining pErk1/2 in cholinergic interneurons) and Macaques (for testing SAG). If the use of distinct models than the AK^{-/-} mice should be acknowledged, the conclusions drawn from these experiments are quite abusive and should be minimized.

For instance, it is written in the Abstract that "We demonstrate that augmenting Shh signaling with agonists of the Ssh effector Smoothens attenuates LID in mouse and macaque models of PD". (only SAG has been tested in monkey).

Also, in the Result part (conclusion of the first chapter): together, "These results from mouse and macaques ...", "macaques" should be removed since just a subset of the experiments have been performed in this case (SAG pharmacology).

These sentences can mislead the reader since experiments conducted in macaques have been performed solely for SAG pharmacology. Please revise such statements in the Abstract, Introduction and Discussion.

2) Most of the experiments are conducted in AK^{-/-} but TH quantification is only given for 6-OHDA-lesioned mice (extended data Fig. 1). Please provide data (and figure) concerning the rate of DAN cell death in AK^{-/-} mice used in this study.

In the same line, L-Dopa 25 mg/kg were used in AK^{-/-} to induce LID, whereas 5 mg/kg were sufficient in 6-OHDA mice. 5 mg/kg is already a high dose for 6-OHDA-lesioned mice (half dose would have been sufficient), but 25 mg/kg represents a high dose. This makes me wonder whether AK^{-/-} mice is an appropriate murine PD model. Also, it is easier to overcome the deleterious effect of LID in a mouse line which conserve substantial amount of intact DANs than in a rodent PD model for which more than 70% of DAN died. Indeed, Ssh signaling (as nicely shown by the authors) require dopaminergic co-signaling to be efficient to alleviate LID.

It would be important that the authors show some key experiments (and not only some pharmacology) performed in 6-OHDA-lesioned mice. For instance, experiments in Shh(DAN^{-/-}) and Smo(ChAT-cre^{-/-}) mice.

In addition, SAG concentration also differs considerably between 6-OHDA-lesioned and AK^{-/-} mice (1 vs 20 mg/kg). The full experiments are given in the Extended data Fig2, but it should be illustrated in the main figure (and clearly written in the Result part) that with SAG 1 mg/kg (and L-dopa 5 mg/kg), there is no significant effect.

Similar comments for Smo(ChAT-cre^{-/-}) mice where L-Dopa 25 mg/kg (again a huge dose) was used.

3) MPTP-treated Macaques experiments. Only SAG molecule was tested in macaques, and one would expect also PUR testing. Also, it is important to provide the details of the statistics (exact p values) for these experiments Fig1F); Indeed, the information given in the Legend is not sufficient (p<0.05) to judge of the real stats since 4 animals have been used and the effect seems quite light and is not fully convincing. Moreover, in MPTP-treated macaque the DAN death rate is expected to be higher than in AK^{-/-} mice (and in the range of those obtained with 6-OHDA-treated mice), and thus SAG 27 mg/kg is a very high dose.

Again, it is important to strengthen these experiments to be able to firmly conclude about an effect of SAG.

4) Pharmacology SAG and PUR. The authors conclude (Abstract and other section) that agonists (plural) of Smo alleviate LID, but PUR needs to be tested in AK^{-/-} mice (and not only in 6-OHDA-treated mice).

Optimally, PUR should have been tested in macaques to claim that Smo agonists are efficient in murine and non-human primates (as stated by the authors).

5) Result part, second chapter. The authors used SAG 25 mg/kg in Smo(ChAT-cre^{-/-}) mice. Why such dose since in Figure 1 there used 20 mg/kg?

Minor comments:

- 1) Abstract: add "Smo" abbreviation when it is first cited.
- 2) Results text: Fig. 1B and C, there is no "white bars" but grey bars.
- 3) Results: DAN are unable to follow (fire) optogenetic activation at 60 Hz. Why did you choose to apply 60 Hz stimulation?

The authors wish to thank the reviewers for their positive and constructive critique of our manuscript. Please find uploaded a revised version of the manuscript that incorporates:

(1) New data

- (i) Supplementary Figure 1 in response to Reviewers 2 and 3: Quantification and comparison of striatal TH fiber density and numbers of DAN in the midbrain of all paradigms used in the study.
- (ii) Supplementary Figure 5 in response to Reviewer 2: Quantification of pERK in fast spiking interneurons revealing no change in 6-OHDA and *AK*^{-/-} mice following Shh manipulations.
- (iii) Supplementary Table 1 in response to Reviewer 3: Cross-reference of all paradigms with all drug conditions and assays performed for quick reference.
- (iv) Supplementary Table 2: Cross-reference of genotypes for all experimental and control recombinant mouse lines used as well as ages of analysis.

(2) Elevation of previous supplemental data into main Figures

- (i) Figure 1 G, H in response to Reviewer 3
- (ii) Figure 3 in response to Reviewer 2
- (iii) Figure 4 D, H in response to Reviewer 2

(3) Revised data displays following revised statistical analysis in response to Reviewer 1:

- (i) Figure 1 H
- (ii) Figure 2 B C
- (iii) Figure 4 D H
- (iv) Figure 5 G
- (v) Figure 7 B D F

(4) Edits throughout to improve clarity, specificity, and readability in response to Reviewers 2 and 3

(5) Updated discussion in response to Reviewer 2

(6) Updated references

Please find below a point-by-point response to each concern and suggestion of the reviewers.

Reviewer #1:

Cholinergic interneurons (CIN) in the dorsal striatum are implicated in L-Dopa induced dyskinesia (LID), however the precise mechanisms underlying LID are unclear. In this manuscript, Malave and colleagues show that LID can be attenuated by increasing smoothed signaling in 6-OHDA or aphakia KO (*AK*^{-/-}) mice. They also report a modest decrease in Dyskinesia Score in MPTP-exposed macaques following treatment with 27 mg/kg of SAG. In LID model mice, inhibition of smoothed signaling with cyclopamine increased the number of CIN with positive pERK1/2 staining, whereas SAG treatment had the inverse effect. The authors show that conditional genetic ablation of Smo from CIN exacerbates abnormal involuntary movements (AIMs) in L-dopa treated mice. By contrast, constitutive activation of Smo signaling, specifically in CIN, prevents AIMs in *AK*^{-/-} mice. Upstream of these events, Malave and

colleagues show ablation of Shh from dopaminergic neurons increases AIMs in mice. Similarly, depleting Shh signaling from dopaminergic neurons through prolonged stimulation increases AIMs. The authors also report that Shh Signaling from dopaminergic neurons is necessary for proper CIN physiology, as determined by loss of phospho-rpS6 intensity in CIN upon dopaminergic Shh ablation. Altogether, their study convincingly shows that the onset of AIMs in mice can be attenuated by increasing sonic hedgehog signaling in CIN.

Author Response: The authors thank the reviewer for their enthusiastic review of our manuscript.

Reviewer #1:

Publication of this interesting and important manuscript is strongly supported after the following points have been addressed:

1. Please explain the rationale for reporting total AIM score in Figure 1B versus AIM duration in Figure 1C. Is duration included in the methods?

Author Response: Figure 1B quantifies LID in a unilateral dopamine depletion model (“6-OHDA”-model) and Figure 1C quantifies LID in a genetic dopamine depletion model (“aphakia”-model). While these models have different etiologies, both are thought to have face- (axial or symmetric dyskinetic movements), construct- (dependent on previously established DA hypersensitivity and repeated L-Dopa dosing) and predictive- (drugs used in the management of LID in humans attenuate dyskinetic movements in the model) validity. A recent and possibly most comprehensive discussion of these models can be found in [1].

The unilateral application of the neurotoxin 6-OHDA results in unilateral hypo-dopaminergia and in lateralization of locomotor output with associated rotational and torsional aberrant movements. In contrast, in the genetic aphakia model the mutations in the Pitx3 gene results in bilateral and symmetric dopaminergic denervation of the striatum and hypodopaminergia. The different etiologies of these models require model specific test conditions and read-outs. We followed for each model well-established and specific quantification protocols, which were validated previously. These scoring methods report either a **composite score of the number of** occurrences of different types of distorted, unilateral movement episodes in the 6-OHDA model or the overall **duration** of “2 or 3 paw” shaking and sliding in AK mice.

We have updated the method section for scoring AIMs in 6-OHDA and AK^{-/-} mice and included there and in the main text the references for the validation of these scoring schemes.

Changes in Manuscript:

A) Methods:

(Lines 1260 to 1282) Abnormal Involuntary Movements (AIMs) Scoring in Mice

6-OHDA lesion model

For the unilateral 6-OHDA lesion model, AIMs score was assayed on the established behavioral rating scales (Cenci et al.,2002, Cenci et al., 2007, Lundblad et al., 2002). Briefly, there are four types of AIMs: 1) abnormal rotational locomotion, 2) axial AIMs, showing dystonic posturing and severe twisting of the head or neck, 3) limb movements with rapid, jerky movements of the front or hind limbs, and 4) orofacial movements of abnormal chewing, licking, grooming, or sticking out of the tongue. The four types of AIMs were scored on a severity scale of 0-4, with 0 exhibiting no abnormality and 4 showing uninterrupted dyskinetic movement. 1-2 blinded experimenters rated AIMs. AIM scores were assessed 35 minutes post treatment injection and 20 min post L-Dopa

injection. Animals were single caged during AIM assessments. Scores for each animal were recorded every 20 minutes, for 1 min each, over 2 hours. The total sum of all four AIM types scores throughout 2 hours was calculated as the “Total AIM” score.

AK^{-/-} mouse model

AK^{-/-} AIMs score was rated on a time scale similar to that described in Ding et al., (2008). These AIMs scores were acquired differently compared to 6-OHDA lesion model because *AK^{-/-}* mice have a bilateral reduction in dopaminergic innervation of the striatum and thus do not exhibit unilateral turning behavior. Instead, *AK^{-/-}* mice were recorded in a clear plastic cylinder (10cm in diameter and 12.5cm in height) 30 minutes following either saline (baseline day) or L-Dopa (test day) injection. Mirrors were placed behind the cylinder to allow for ventral views of the behavior. Each trial was five minutes with a 30-second habituation period followed by behavioral scoring. The “total AIM duration” was quantified for each trial as the length of time the animal displayed sliding or shaking of the forelimb paws while rearing on the cylinder wall. “Three-paw” dyskinesias were also recognized as events with shaking of two forelimbs and one hindlimb. “Four-paw” dyskinesias were observed in very extreme dyskinetic cases when the animal balanced on its tail for brief moments Ding et al., (2008).

B) Results:

(Lines 107 to 115) We first tested whether chronic augmentation of Shh signaling throughout L-Dopa therapy could attenuate the development of dyskinesia in preclinical models of LID with diverse etiology. Model-specific behavioral tests with predictive validity for examining LID following genetic or neurotoxin-induced degeneration of DAN in rodents and non-human primates have been previously developed [1]. Prior work utilizing these behavioral measures has demonstrated that LID develops in an L-Dopa dose-dependent manner that is model specific across different species and forms of induced DAN degeneration [1]. We quantified LID in these animals by a subjective measure called the abnormal involuntary movement (AIM) scale which captures model specific features of LID [1-3].

(Lines 128 to 133) As has been previously shown, daily injection of *AK^{-/-}* mice with L-Dopa at 25 mg/kg resulted in presentation of *AK^{-/-}* model-specific AIMs quantified as the total amount of time that the animal spends sliding 3 paws along the walls of a glass cylinder while rearing on one leg [4]

- 2. The decrease in dyskinesia score in macaques following SAG treatment is not as convincing as the results shown for mice—why was a different statistical test used and please discuss why response may have been different**

Author Response: Attrition of an anti-dyskinetic effect is a well-known phenomenon in the LID field [5] [6] where manipulations in rodents exhibit a larger magnitude of effect than observed in non-human primate models.

For the rodent studies, we relied on the numerous empirical SAG dose selections described in the literature. Such data are surprisingly lacking for non-human primates despite SAG being a well-studied molecule [7]. Literature searches failed to identify useful pharmacokinetics / pharmacodynamics studies. No primary data are available to the best of our knowledge and most reviews focus upon the in vitro data [8, 9]. Not knowing the pharmacokinetics or the target engagement dose in primates, we adopted a careful dose-escalating regiment. The range of SAG dosing was guided by our finding in mice that the dose of SAG needed to attenuate LID scales with the dose of L-Dopa used to induce dyskinesia. Consistent with the L-Dopa dose of 18 to 22 mg/kg of L-Dopa in Macaques (Methods), we expected only the 27 mg/kg SAG dose to be

effective. The results confirmed this prediction. Due to limitations in resources we were unable to test higher doses or to increase the N of subjects.

Further, non-human primate behavior was rated as per most recent guidelines [10] which include the recommendation to use non-parametric tests (nature of the rating scale and number of individuals). Non-parametric analysis is less sensitive compared to parametric analysis and reduces the risk of identifying false-positive responses. We recognized that we did not state the full statistical analysis. We therefore amended the figure legend.

Changes in Manuscript:

A) Methods:

(Lines 1293 to 1295) Once parkinsonism was stable, levodopa (Madopar®, Roche, Levodopa/carbidopa, ratio 4:1) was administered twice daily for 4-5 months at an individually-tailored dose (18 to 22 mg/kg) designed to produce a full reversal of the parkinsonian condition (p.o. by gavage).

(Lines 1314 to 1317) Primate behavior was rated as per most recent guidelines [10] which include the recommendation to use non-parametric tests (nature of the rating scale and number of individuals). Non-parametric analysis is less sensitive compared to parametric analysis and reduces the risk of identifying false-positive responses.

B) Results:

(Line 159 to 159) "...showed reduced dyskinesia..." to "showed a modest but significant reduction in dyskinesia"

(Line 162): deleted: "These results suggest that acute activation of the Shh/Smo pathway in non human primate models of LID is capable attenuating AIMs..."

C) Figure 1 Legend:

(Lines 686 to 689) Dyskinesia score in dyskinetic Macaques was reduced in response to a within-subject drug escalation of SAG at the highest dose (0, 3, 9, and 27 mg/kg; n=4; (Friedman's nonparametric RM one-way ANOVA, $F= 10.89$, $P= 0.0020$, followed by Dunn's post hoc test, $P= 0.0185$).

- 3. Cholinergic interneuron pERK1/2 staining is scored as either positive or negative in Figure 1H-I and Extended Figure 5. If possible, the authors should report normalized pERK1/2 fluorescence intensity in CIN, analogous to their measurement of phospho-pS6 intensity in Figure 5.**

Author Response: We agree and thank the reviewer for this suggestion. We report all pERK1/2 data as normalized fluorescence intensity in CIN in this revision.

Changes in Manuscript:

A) Methods

(Line 1364 to 1365) NeuN fluorescence intensity of the same cell was used for normalization of fluorescence across treatment and genotype groups.

- 4. Data points in Figure 1H-I, Extended Figure 5, and Figure 5 B,D,F, are displayed as total cells instead of animals per genotype. Our statistical experts tell us that they should be reported by animal and displayed by animal rather than total cells counted.**

Author Response: We agree and thank the reviewer for bringing this issue to our attention. All histological quantification data (pERK and p-rpS6) is now displayed by animal. All statistical analysis have been re-run accordingly.

Changes in Manuscript:

A) Methods:

(Line 1369): Each data point reflects the average percent of pErk^{1/2} in one animal

(Line 1378): Each data point reflects the average level of p-rpS6 intensity in one animal.

B) Figures:

Figure 2 B and C.

Figure 4 D and H.

Figure 5 G

Figure 7 B, D, F

Supplementary Figure 1

Supplementary Figure 5

- 5. If possible, the authors should determine if LID attenuation by Smo signaling occurs via the canonical Hedgehog pathway or by another pathway. For example, is Gli1 or another classical pathway marker changed in CIN following SAG treatment, decreased in CIN by cyclopamine, or relatively decreased in AK-/- mice when compared to non-transgenic littermates?**

Author Response: We thank the reviewer for these suggestions. It was found previously that striatal injections of SAG and Cyclopamine effect the transcription of Gli3 and muscarinic autoreceptor M2 in a dose dependent manner [11]. However, in contrast to those gene expression changes that became apparent after 36 hours, in the current study we find that SAG modulates the intensity of LID within minutes. This timeframe excludes a transcriptional based mechanism and instead suggests a transcription independent, more immediate signaling pathway downstream of Smo that impinges on LID expression. Smo is well known to couple to Gi and to promote rapid recruitment of 2nd messengers in addition to effecting gene expression at greater time scales [12-14].

Further, the pharmacological attenuation of the LID phenotype in mice with conditional ablation of Shh from DAN reveals that the long-term absence of Shh/Smo signaling does not erode the signaling pathway by which acute SAG activation of Smo influences LID expression. Importantly, the conditional ablation of Smo from CIN blocks the attenuation of LID by SAG. Together these observations reveal that Shh/Smo dependent modulation of LID occurs through a CIN resident mechanism that must involve the transcription independent arm of Smo signaling.

Changes in Manuscript:

A) Introduction:

(Lines 96 to 98): Smo, a G α_i -coupled GPCR, is capable of impinging on neuronal intracellular Ca⁺⁺ levels within minutes via G-protein signaling as well as regulating target gene expression within hours via activation of Gli transcription factors [12-14].

B) Discussion:

(Lines 409 on): Shh/Smo signaling is known to act across multiple time scales via both rapid G-protein-dependent signaling and slower transcriptional regulation of target genes [15, 16]. The activation of these diverse signaling pathways suggests that Shh_{DAN} to Smo_{CIN} signaling might impinge on several homeostatic mechanisms which help maintain physiological CIN activity and which in turn become dysregulated following DAN degeneration and the loss of Shh_{DAN}. The

potential action of these downstream signaling cascades is supported by several of our observations in the current experiments. For example, we observed that acute Smo agonist exposure can reduce pERK and boost p-rpS6 levels in CIN within minutes through a mechanism that requires Smo expression on CIN.

(Lines 441 on): Interestingly, SK channels are inhibited by muscarinic M2/M4 autoreceptor signaling [17-19] which is increased in the absence of Shh_{DAN} to Smo_{CIN} signaling [11]. Thus, reduced Smo_{CIN} activity following the loss of Shh_{DAN} could depress SK channel activity and cause, in part, the altered electrophysiology observed in CIN following DAN degeneration and L-Dopa administration [20].

Reviewer #1 Minor points:

1) Define M4PAM in the main text.

Author Response:

(Line 214): It was previously shown that muscarinic acetylcholine receptor M4 positive allosteric modulators (M4-PAM) restores long-term depression in DR1-MSN of the striatal direct output pathway and attenuates LID [21, 22].

(Line 210 to 229): M4PAM experiment results have been elevated to Figure 3 in main text.

2) Use consistent abbreviation for cholinergic (ACh) throughout the text.

Author Response: we refer to cholinergic interneurons as “CIN” and to acetylcholine as “ACh” throughout the manuscript.

3) Extended data Fig2C—is a linear R2 response expected when the x axis is nonlinear?

Author Response: Thank you for raising this point. We consulted with enzyme biochemists and we understand now that calculating R2 is an over-analysis of the data. R2 was removed from the panel. The word “linearly” was struck from the result section.

Changes in Manuscript:

(A) Results:

(Line 143) “linearly” deleted.

(B) Figures:

Figure 1 H: R2 line and value removed from panel

(C) Figure legends:

Figure Legend 1 G, H, I: R2 calculation and mention removed.

Reviewer #2:

Malave and colleagues present a novel, substantial evidence for the role of cholinergic interneurons of the striatum in the pathophysiology of levodopa-induced dyskinesia (LIDs), a disabling complication of long-term dopaminergic therapy commonly observed in parkinsonian patients. This elegant, multidisciplinary work comes from an excellent group of scientists, experts in the field, which makes this paper relevant and timely, and represents a fundamental step for clarifying the pathophysiology of Parkinson’s disease. By shedding new light on the role of SSh signalling pathways, a previously underestimated mechanism, this work has a potential translational relevance and could open the way to novel pharmacological treatments.

The authors provide an accurate description of the role of Sonic Hedgehog (Ssh) co-signaling with dopamine in dopaminergic target areas, specifically the striatum. They show that reduced Shh signaling in

dopamine-depleted animals induces expression of LID, whereas restoring the balance of dopamine and Shh signaling by agonists of the Shh signaling effector receptor (Smo) reduced dyskinesias caused by repeated levodopa treatment both in mouse and macaque models of Parkinson's disease. Yet, expression of constitutively active Smo (SmoM2) in cholinergic interneurons induces a resistance to develop LIDs in the aphakia model of PD.

By utilizing three well-established PD animal models, the authors developed novel genetic and optical stimulation paradigms for manipulating the strength of dopamine and Shh co-transmission by dopaminergic cells onto striatal cholinergic interneurons in the manifestation of dyskinesias. In both animal species, they observed that dyskinesias are caused by a reduction of Shh-dopamine/Smo signaling on cholinergic targets, and conversely, that enhancing Smo signaling in these interneurons attenuates the involuntary movements through a mechanism that might involve a direct modulation of the striatal output pathways. Phosphorylation of Erk in the striatum is considered a LID marker. Modulation with Smo agonists in cholinergic interneurons was able to modify the occurrence of LIDs. Of note, neither the chronic nor the acute dosing with SSh agents (SAG) affected the prokinetic efficacy of levodopa.

Author Response: The authors thank the reviewer for their enthusiastic review of our manuscript.

Reviewer #2: There are some points that require clarification, that would strengthen the significance of these findings.

- 1. Clinical evidence does not support the efficacy of anticholinergic drugs in the management of dyskinesia. A relevant question is whether anticholinergic worsen dyskinesias. The authors use M4 PAM which is hypothesized to work on MSN (Fig 7). However, M4 autoreceptor activation on cholinergic interneurons is a well established mechanism of autoinhibition and expected to cause a decrease in ACh release, and thus of cholinergic tone. This should be taken into consideration.**

Author Response: Thank you for this insightful comment. It was previously found that M4 PAM attenuated LID and LID associated aberrant plasticity in mice and non-human primates [21]. That study found that M4 PAM acts directly on D1R MSNs where it prevented aberrant plasticity. More recently, consistent with the aforementioned study by Shen et al 2016 and our own findings, Brugnoli et al., 2020 observed that striatal as well as nigral M4 PAM attenuates LID [22]. In all three studies, M4PAM treatment should also lead to a decrease of cholinergic tone due to auto-receptor activation. In turn, decreased ACh tone should facilitate LID according to our own data. However, since the LID attenuating function of M4 PAM resides post-synaptic to the auto-receptor activity of M4PAM, the effect on post synaptic D1R MSNs overrides the reduction in ACh tone via M4 PAM's action on auto-receptors.

We have revised Figure 3, revised text in results and discussion, and have updated the bibliography with Brugnoli et al., 2020 in order to visualize and to clarify the hierarchical effects of M4PAM regulation of cholinergic tone and DR1-MSN activity.

Change in Manuscript:

(A) Results:

(Line 210 on): Title of chapter: Smo Modulates AIMs via a Mechanism Upstream of Cholinergic Influence on Striatal Output

(Line 216 to 217): Muscarinic acetylcholine receptor M4 also acts as an inhibitory autoreceptor on CIN.

(B) Figures:

Figure 3 A: Scheme indicates auto receptor and post-synaptic action of M4PAM onto CIN and DR1 MSN, resp.;

(C) Discussion:

Line 380 to 384: Thus, our data indicate that signaling from Shh_{DAN} to Smo_{CIN} prevents LID formation and expression through a mechanism that impinges on CIN physiology, and thus acts upstream of previously reported LID-attenuating mechanisms such as muscarinic-dependent LTD of D1R MSNs [21] or possible nicotinic receptor desensitization [23].

2. The role of nicotinic receptors that are known to promote DA release should be included in the proposed scenario (Threlfell et al 2012, Neuron; Brimblecombe et al., 2018 eNeuro).

Author Response: We agree. An additional mechanism by which elevated ACh tone could attenuate LID is through desensitization of presynaptic nicotinic receptors resulting in the dampening of DA release onto D1 MSN. Evidence for such a mechanism in regard of LID modulation was provided by Quik et al., 2013 a, 2013 b [24, 25]. Indeed, these findings further support our findings that increasing ACh tone and stimulation of CIN by Shh/Smo signaling have an anti-dyskinetic effect.

Changes in manuscript:

(A) Results:

Lines 226 – 228:

These findings are consistent with the possibility that Smo-mediated modulation of AIMs occurs upstream of M4PAM dependent attenuation of AIMs, and therefore presynaptic to DR1-MSN inputs from CIN.

Lines 287 – 290:

Thus, the suppression of LID by Shh_{DAN} via Smo_{CIN} occurs likely antecedent to LID inhibition by muscarinic signaling on D1 MSN via M4PAM ([21], Fig. 3), or nicotinic receptor desensitization proposed to occur upon optogenetic CIN activation [23].

(B) Discussion:

Line 380 to 384:

Thus, our data indicate that signaling from Shh_{DAN} to Smo_{CIN} prevents LID formation and expression through a mechanism that impinges on CIN physiology, and thus acts upstream of previously reported LID-attenuating mechanisms such as muscarinic-dependent LTD of D1R MSNs [21] or possible nicotinic receptor desensitization [23].

3. Is there a specific effect for striatal targets? Does any other DAN target region express SSH-coupled receptors? Did the authors observe such specificity?

Author Response: Thank you for raising the important point of cellular specificity of Shh signaling in the striatum. We found previously that only CIN and Parv⁺ fast spiking interneurons (FSN) express the Shh signaling machinery (i.e. Ptc1 and Smo) [11]. In new data, we show here in the context of LID that Smo pharmacology affects the phosphorylation state of ERK only in CIN but not in FSN. Further, the effects of Smo pharmacology in CIN are more pronounced in the DLS than in the DMS. In this revision, we have included FSN data and contrasted the results with CIN for both the 6-OHDA and AK^{-/-} paradigms.

Changes in Manuscript:

(A) Results:

(Line 199 to 202): Parvalbumin expressing fast spiking interneurons (FSN) of the striatum are also sensitive to Shh signaling from DAN and express Ptc1 and Smo [11]. However, Smo agonist treatment did not alter the phosphorylation status of pERK in FSN

among of L-Dopa treated 6-OHDA or AK^{-/-} animals (supplementary Fig. 5).

(B) Figures:

Supplementary Figure 5 A – C.

(C) Figure Legend:

Supplementary Figure 5 A – C.

4. The authors claim that ERK changes are observed in cholinergic cells. However there is abundant literature showing that spiny neurons are preferentially affected, in particular those expressing D1 receptors (Cholinergic interneurons express D5 receptors). Indeed, spiny neurons of the direct pathway display a dramatic increase in cyclic AMP-responsive element binding (CREB), AP-1, and ERK-dependent gene expression (Heiman et al., 2014 PNAS).

Author Response: Agreed. We recognize that we did not clearly explain the context and rationale for specifically quantifying pERK in CIN. Thank you for bringing this to our attention.

There are several reasons for our focus on ERK phosphorylation in CIN. (1) While the phosphorylation of ERK in MSN is widespread and immediate [1], ERK phosphorylation in CIN of the dorsolateral striatum becomes more prominent with repeated dosing [26]. (2) The dorsolateral region of the striatum (DLS) harbors neurons critical for LID expression. Specifically, stimulation of “activity trapped neurons” in this area that were active during LID has shown to elicit AIMS [27]. (3) In contrast to MSN, CIN express the Shh receptor Ptc and effector Smo [11] and thus the phosphorylation state of ERK in MSN must be an indirect and downstream effect of Shh signaling to CIN while ERK phosphorylation in CIN could be a direct target of Shh signaling. Together, these observations pointed to the possibility that ERK phosphorylation selectively in CIN of the DLS could be critical for LID and be regulated by Smo signaling. Our data supports this hypothesis since we show that Smo agonists modulate ERK phosphorylation selectively in CIN of the DLS but not DMS which correlates with the severity of AIMS in the utilized LID paradigms.

In the revised manuscript, we introduce the section about pERK in CIN by stating and referencing the finding of pERK in D1R MSN in response to L-Dopa treatment. We then continue by pointing out that repeated L-Dopa dosing results in ERK phosphorylation among CIN of the DLS, which correlates with LID severity.

Changes in manuscript:

(A) Results:

(Line 187 to 192): During early LID, pErk^{1/2} (phospho-Thr202/Tyr204–ERK^{1/2}) is observed in a dispersed pattern within the striatum among primarily medium spiny projection neurons (MSN) expressing the dopamine D1 receptor (DR1) [28]. However, dosing with L-Dopa progressively increases pErk^{1/2} prevalence among CIN in the dorsolateral striatum (DLS) in correlation with increased LID severity [29].

5. **Prolonged optogenetic activation is expected to potentially deplete other neurotransmitters as well. Did the authors rule out this possibility?**

Author Response: We cannot rule out an effect on other neurotransmitters. However, using Smo pharmacology, we provide positive evidence for a depletion of Shh in correlation with increased AIMS. Consistent with the significance of that depletion for the expression of AIMS, we find that Smo agonist treatment attenuates the stimulation-induced expression of AIMS.

Reviewer #2: Minor points.

- a) some typos throughout the text, please amend.

Author Response: manuscript was proof read by several native English speakers and multiple typos were corrected.

- b) -“it appears the un-physiological CIN activity which emerges in LID following dopaminergic denervation bestows a LID-inductive function onto CIN which can be disengaged through their ablation.” Please rephrase to simplify this sentence.

Author Response:

Changes to manuscript:

(A) Discussion:

(Line 401 – 408): Ablation of CIN, which has been shown to block LID [26], might thus prevent the formation of a LID facilitating gain of function of CIN in absence of Shh_{DAN} to Smo_{CIN} signaling. Indeed, our results point to a close, inhibitory cross talk between dopaminergic and Shh signaling in CIN (Fig. 8 E) and suggest that the interruption of Shh_{DAN} to Smo_{CIN} signaling is not merely permissive for LID but is a critical mechanism by which LID-inducing CIN pathology is set in motion following the loss of DAN. A parsimonious possibility is that DA signaling on CIN unrestrained by Shh_{DAN} to Smo_{CIN} signaling is a key facilitator of LID.

Reviewer #3:

In this study, Malave and colleagues examine the impact of sonic hedgehog (shh) signaling in the dorsal striatum (with distinguishing dorsolateral and dorsomedial striatum) in L-Dopa induced dyskinesia (LID) animal models. The authors show that agonists (SAG and PUR) of the shh effector Smoothed (Smo) decrease the severity of LIDs. In line with this, they show (with elegant transgenic approaches) that decreasing and increasing ssh signaling (acting on the source, the dopaminergic neurons, or the target, the striatal cholinergic interneurons) allow to increase and decrease, respectively, the severity of LID. Overall, this constitutes an impressive amount of work, providing in-depth analysis of a crucial problem for parkinsonian patients: how to alleviate the LID (besides DBS which is a successful symptomatic therapy but a very invasive procedure). The data are very interesting and novel.

Author Response: The authors thank the reviewer for their encouraging feedback.

Reviewer #3: Despite a huge body of experiments (notably the impressive use of sophisticated transgenic mouse lines), some questions remain open and some experiments do not fully support the very strong conclusion brought by the authors about the effect of Shh signaling for alleviating LID.

- 1. The authors used in this study a murine model of Parkinson’s disease: the AK-/- mice. In Figure 1 they also used for specific subtypes of experiments another rodent model, the 6-ODHA-lesioned mice (for testing SAG and PUR and examining pErk1/2 in cholinergic interneurons) and Macaques (for testing SAG). If the use of distinct models than the AK-/- mice should be acknowledged, the conclusions drawn from these experiments are quite abusive and should be minimized.**

Author response: The authors thank the reviewer for pointing out that we failed to articulate clearly the utility of using each animal model.

There is currently no single animal model that recapitulates all features of Parkinson's or LID. Instead, it is assumed that different models reflect a subset of pathological features. Thus, by using two models that differ significantly in their etiology (6-OHDA and *AK*^{-/-}), we aimed to mitigate the shortcomings of any one model and the possibility of inadvertently mistaking an idiosyncratic feature of a particular model for LID.

Please note that we tested in both models whether Smo modulates the severity of model-specific AIMs in a dose-specific manner upon chronic or acute dosing. We also examined the phosphorylation status of ERK in CIN of the DLS and DMS across both models. Further, in both models, we revealed that the dose of Smo agonist SAG needed for attenuation of AIMs scaled with the dose of L-Dopa used to induce dyskinesia. For the 6-OHDA model we extended these observations to a 2nd agonist of Smo (Purmorphamine) and additionally showed that the effects of SAG are rapidly reversible.

For a further discussion of the utility and validation of these models, please also refer to our response to reviewer 1 point 1.

Throughout the manuscript, we added qualifiers detailing the association between models and test. In order to allow the reader to appreciate the utility of the comparative analysis of all animal paradigms we appended a supplementary table that lists paradigms, conditions, and assays performed throughout the manuscript.

Changes in manuscript:

(A) New supplementary table 1:

Supplemental Table 1 lists paradigms, conditions and assays performed.

(B) Results:

(Line 109 to 117): We first tested whether chronic augmentation of Shh signaling throughout L-Dopa therapy could attenuate the development of dyskinesia in preclinical models of LID with diverse etiology. Model-specific behavioral tests with predictive validity for examining LID following genetic or neurotoxin-induced degeneration of DAN in rodents and non-human primates have been previously developed [1]. Prior work utilizing these behavioral measures has demonstrated that LID develops in an L-Dopa dose-dependent manner that is model specific across different species and forms of induced DAN degeneration [1]. We quantified LID in these animals by a subjective measure called the abnormal involuntary movement (AIM) scale which captures model specific features of LID [1-3].

(Line 117): In the 6-OHDA rodent model....

(Line 125): We next tested the effect of Smo stimulation and inhibition on AIMs in the aphakia (*AK*^{-/-}) mouse line.

(Line 143 to 145): The ability of SAG to reduce AIMs in *AK*^{-/-} mice was similar in magnitude to the effect observed following administration of the anti-dyskinetic bench mark drug amantadine (60 mg/kg; [30]; Fig. 1 E, green bar).

(Line 170): This study revealed that the dose of SAG needed to attenuate LID by 50 % in each of the L-Dopa groups scaled with the dose of L-Dopa used to induce dyskinesia (Fig. 1 H and I).

(Line 17 to 176): Concordant with the L-Dopa to SAG dose relationship in *AK*^{-/-} mice, we observed in the 6-OHDA paradigm that greater doses of SAG (20 mg/kg) were required to attenuate AIMs after L-Dopa was ramped up from 5 to 20 mg/kg. Conversely, AIMs produced by

lower L-Dopa doses in the same animals (5 mg/kg) could be attenuated by treatment with lower doses of SAG (10 mg/kg) (supplementary Fig. 4).

(Line 178 to 180): Together, these findings indicate that in a neurotoxin and genetic model of LID, the momentary relative imbalance between Shh and DA signaling determines the severity of LID.

- 2. For instance, it is written in the Abstract that “We demonstrate that augmenting Shh signaling with agonists of the Ssh effector Smoothened attenuates LID in mouse and macaque models of PD”. (only SAG has been tested in monkey).**

Author Response: Agreed.

Changes to manuscript:

(A) Abstract:

(Line 47 to 49): changed “We demonstrate that augmenting Shh signaling with agonists of the Shh effector smoothened (Smo) attenuates LID in mouse and macaque models of PD” to “We find that the pharmacological activation of the Shh effector Smoothened (Smo) attenuates LID in the neurotoxic 6-OHDA- and genetic aphakia mouse models of PD.

- 3. Also, in the Result part (conclusion of the first chapter): together, “These results from mouse and macaques ...”, “macaques” should be removed since just a subset of the experiments have been performed in this case (SAG pharmacology).**

Author response: Agreed.

Change to manuscript:

(A) Results:

(Line 200): changed “..these results from mouse and macaque models of LID indicate that..” to “these results indicate...”

- 4. These sentences can mislead the reader since experiments conducted in macaques have been performed solely for SAG pharmacology. Please revise such statements in the Abstract, Introduction and Discussion.**

Authors Response: Agreed.

Changes to manuscript:

(A) Abstract:

(Line 47 to 49): changed “We demonstrate that augmenting Shh signaling with agonists of the Shh effector smoothened (Smo) attenuates LID in mouse and macaque models of PD” to “We find that the pharmacological activation of the Shh effector Smoothened (Smo) attenuates LID in the neurotoxic 6-OHDA- and genetic aphakia mouse models of PD.

(B) Results:

Please see above response to point 1.

Introduction:

No mention.

Discussion:

No mention.

- 5. Most of the experiments are conducted in AK-/- but TH quantification is only given for 6-OHDA-lesioned mice (extended data Fig. 1). Please provide data (and figure) concerning the rate of DAN cell death in AK-/- mice used in this study.**

Author response: The authors thank the reviewer for raising this critical issue and for pointing out that we failed to adequately explain the changes to DAN that underpin the 6-OHDA, MPTP and *AK*^{-/-} models of LID. We thank the reviewer for this suggestion.

Changes to manuscript:

(A) Figures:

supplementary Figure 1.

supplementary Figure legend 1: TH fiber density or numbers of Midbrain DAN do not predict LID formation and expression. **(A)** Tyrosine hydroxylase (TH) fiber density in the dorsolateral striatum (DLS, dotted quarter circle) was visualized as a proxy for the integrity of dopaminergic projections across the mouse models utilized in this study. **(B)** For quantification of Striatal TH fiber density, TH staining intensity was normalized to background signal and average TH intensity in the DLS per animal was reported. Results are reported as percent difference between experimental animals and average TH intensity of control animals (n = 8–12, 3-5 planes each; unpaired two-tailed student's t test **** P<0.0001 6-OHDA lesioned hemisphere vs. 6-OHDA unlesioned hemisphere, *AK*^{-/-} vs *WT*, *AK*^{-/-} *SmoM2*^{CIN^{+/-}} vs *WT*, *Smo*^{CIN^{-/-}} vs *Smo*^{CIN^{+/-}}; **** P<0.001 *Shh*^{DAN^{-/-}} vs *Shh*^{DAN^{+/-}}. All bar graphs are plotted as mean +/- SEM. **(C)** DAN cell bodies were identified as TH positive cell soma on coronal sections of the mesencephalon in all mouse models used in this study. **(D)** For quantification of Midbrain DAN Soma, TH background signal was set as a threshold and all soma in the VTA and SNpc with TH signal above background were counted. Results are reported as percent difference in number of TH+ soma between experimental animals and control animals (n = 8–12, 3-5 planes each; unpaired two-tailed student's t test ** P<0.01 6-OHDA lesioned hemisphere vs. 6-OHDA unlesioned hemisphere, *AK*^{-/-} *SmoM2*^{CIN^{+/-}} vs *WT*, * P<0.05 *AK*^{-/-} vs *WT*. All bar graphs are plotted as mean +/- SEM.

(B) Results:

(Line 115 to 120): In the 6-OHDA rodent model, unilateral striatal injections of the dopaminergic toxin 6-OHDA results in dopaminergic denervation of the ipsilateral dorsal striatum, DAN degeneration (comparative quantification of dopaminergic innervation of the striatum and remaining DAN numbers in the midbrain is given for all mouse paradigms used in this study in supplementary Fig. 1) and a rotational bias of locomotion ([3] and supplementary Fig. 2).

(Line 126-128): In *AK*^{-/-} mice, the absence of the transcription factor Pitx3 during development results in a bilateral, severe diminishment but not complete absence of dopaminergic innervation of the dorsal striatum and hypodopaminergia in the adult brain ([31, 32]; quantified in supplementary Fig. 1).

(Line 238 to 240): These mice showed no evidence for dopaminergic fiber density atrophy in the striatum and possess a full complement of DAN at 3 months of age (quantified in supplemental Fig. 1).

(Line 252 to 255): Comparative quantification of striatal dopaminergic fiber density and numbers of DAN in the midbrain at 3 months of age revealed that selective expression of *SmoM2* in CIN did not alter the severe diminishment of the dopaminergic system present in the parental *AK*^{-/-} mouse line (quantified in supplemental Fig. 1).

(Line 269 to 271): *Shh*^{DAN^{-/-}} mice go through a phase of exuberant sprouting of dopaminergic fibers leading to a ~38 +/- 6.8 % increase in dopaminergic fiber density but normal numbers of DAN as young adults (quantified in [11] and in supplemental Fig. 1).

6. In the same line, L-Dopa 25 mg/kg were used in *AK*^{-/-} to induce LID, whereas 5 mg/kg were sufficient in 6-OHDA mice. 5 mg/kg is already a high dose for 6-OHDA-lesioned mice

(half dose would have been sufficient), but 25 mg/kg represents a high dose. This makes me wonder whether AK^{-/-} mice is an appropriate murine PD model.

Author response: Test conditions and read outs differ among each LID model that we used. (Please refer to author response to reviewer 1 point 1). Most likely, this is a reflection of the very different etiology that underlies each model. Yet, each model chosen is thought to have face- (axial or symmetric dyskinetic movements), construct- (dependent on previously established DA hypersensitivity and repeated L-Dopa dosing) and predictive- (drugs used in the management of LID in humans attenuate dyskinetic movements) validity.

Further, it is well-established that the degree of LID that can be observed in any given model and in humans is dependent on (1) the L-dopa dose and (2) the degree of DAN degeneration and corresponding dopamine hypersensitivity. In the animal models, it might also be of importance whether the hypodopaminergia is induced semi acutely by neurotoxins (as in the 6-OHDA model) or is present from development onwards offering greater opportunities for adaptive processes to take hold (as in the genetic model). Further, in regard to the 6-OHDA paradigm, there are two common routes used for application of the neurotoxin: a) injection into the medial forebrain bundle (MFB), or b) injection into the dorsolateral striatum.

Because of the different etiologies that establish the underlying conditions in these models (ie. neurotoxin and different sites of administration or genetic ablation) the type of AIMs that can be observed in each model differ (i.e. unilateral disturbances in the 6-OHDA model vs. bilateral disturbances observed in the genetic AK^{-/-} model). These issues have been discussed in multiple publications. Possibly the most comprehensive and current discussion can be found in Bastide et al., 2015 [1].

Thus, for our efforts to investigate whether Shh signaling plays a common role in diverse models of LID, it was important we use validated protocols for concentration and numbers of L-dopa doses as well as for scoring of AIMs (see amended and further detailed method section, response to reviewer 1 point 1).

In regard to the L-Dopa doses used, we indeed aimed on using the maximal dosing supported by the literature for each paradigm. The declared logic here was to ensure maximal formation and expression of LID in order to create robust assay settings in which we could stringently test if Smo activation would attenuate those LID.

In order to facilitate appreciation of the use of model specific conditions and assays, we added a supplementary table for comparative cross-reference across all paradigms used. We also emphasized throughout the text that drug conditions are specific for each paradigm and based on published assay validation literature with references given:

Change to manuscript

(A) Figures:

Supplementary Table 1: cross-references all paradigms with conditions, drug schedules and doses, and assays.

(B) Results:

(Line 111): Using these behavioral outcome measures demonstrated that LID develops across different species and forms of induced DAN degeneration in an L-Dopa dose-dependent manner [1].

(Line 119): As previously established [2], daily injections of L-Dopa (5 mg/kg) in these animals led to the appearance of unilateral AIMs (aggregate score of axial and orofacial contortions; Fig. 1 B, grey bar).

(Line 126 - 133): In AK^{-/-} mice, absence of the transcription factor Pitx3 during development results in severe bilateral diminishment of dopaminergic innervation to the dorsal striatum and hypodopaminergia in the adult brain ([31, 32]; quantified in supplementary Fig. 1). As has been previously shown, daily injection of AK^{-/-} mice with L-Dopa at 25 mg/kg resulted in presentation

of $AK^{-/-}$ model-specific AIMs quantified as the total amount of time that the animal spends sliding 3 paws along the walls of a glass cylinder while rearing on one leg [4] (Fig. 1 C, grey bar; supplementary Table 1 lists all drug regimens comparatively for all paradigms used in this study).

7. Also, it is easier to overcome the deleterious effect of LID in a mouse line which conserve substantial amount of intact DAN than in a rodent PD model for which more than 70% of DAN died. Indeed, Ssh signaling (as nicely shown by the authors) require dopaminergic co-signaling to be efficient to alleviate LID. It would be important that the authors show some key experiments (and not only some pharmacology) performed in 6-OHDA-lesioned mice. For instance, experiments in $Shh(DAN^{-/-})$ and $Smo(ChAT-cre^{-/-})$ mice.

Author response: The authors thank the reviewer for raising this critical issue and for pointing out that we failed to adequately explain the significance of Shh expression by DAN in regard to the mechanisms that underpin the 6-OHDA, MPTP and $AK^{-/-}$ models of LID.

In order to clearly describe the underlying differences in our paradigms, and as an extension of the suggestion by the reviewer (point 6 above), we quantified TH fiber density and numbers of DAN in all paradigms used (i.e. 6-OHDA, $AK^{-/-}$, $Shh_{DAN^{-/-}}$, $Smo_{CIN^{-/-}}$, $SmoM2_{CIN+/-}$; optogenetically induced). The results further confirm the tenet of our study. Specifically, we see that there exist no correlation between the loss of striatal fiber density or numbers of remaining DAN in the mesencephalon with the expression of LID. In contrast, the commonality among all paradigms in which L-Dopa treatment results in LID is the absence of Shh/Smo signaling. Conversely, the $AK^{-/-}$ animals can be rendered LID resistant by the CIN specific expression of constitutively active SmoM2 without producing greater dopaminergic fiber density or greater numbers of DAN numbers.

Change to manuscript:

(A) Figures:

Supplementary Figure 1.

Supplementary Figure 1 legend: please see response to point # 6.

(B) Results:

Please see response to point # 6.

8. In addition, SAG concentration also differs considerably between 6-OHDA-lesioned and $AK^{-/-}$ mice (1 vs 20 mg/kg).

Author response: One of our critical findings is that the amount of SAG needed to attenuate LID scales with the model-specific dose of L-Dopa used to induce dyskinesia in the 6-OHDA and the $AK^{-/-}$ paradigms. Thus our data suggests an intersection of dopaminergic signaling and Shh/Smo signaling downstream of the receptors on CIN. This conclusion is based on concentration dependent dose response assays, not on single doses. In order to facilitate appreciation of the approach we added a supplementary table that lists drug regimens and underlying assay conditions for each paradigm used.

Changes to manuscript:

(A) Figures:

Supplementary table 1

(B) Results:

(Line128 to 133): As has been previously shown, daily injection of $AK^{-/-}$ mice with L-Dopa at 25 mg/kg resulted in presentation of $AK^{-/-}$ model-specific AIMs quantified as the total amount of time that the animal spends sliding 3 paws along the walls of a glass cylinder while rearing on

one leg [4] (Fig. 1 C, grey bar; supplementary Table 1 lists all drug regiments comparatively for all paradigms used in this study).

9. The full experiments are given in the Extended data Fig2, but it should be illustrated in the main figure (and clearly written in the Result part) that with SAG 1 mg/kg (and L-dopa 5 mg/kg), there is no significant effect.

Author response: Thank you for suggesting that this observation warrants elevation to a main Figure and full description in the main text.

Change to manuscript:

(A) Figures

Figure 1

Figure 1 Legend:

(G) LID were induced in 3 groups of $AK^{-/-}$ mice injected daily for 20 days with either 5, 10 or 30 mg/kg L-Dopa. In probe trials at days 21, 25, 29, and 33, the dose-dependent degree of LID attenuation by SAG was tested using a within-subject escalation strategy including three-day SAG washouts (L-Dopa only) in between probe trials. SAG was serially administered across the probe trials at 4 different concentrations (0.8; 2.5; 7.5; and 15 mg/kg) in each L-Dopa dose group (5, 10 or 30 mg/kg) **(H)** In each L-Dopa concentration group, the attenuation of LID by SAG was SAG dose-dependent. From this data we estimated the SAG concentration that resulted in half-maximal (EC50) inhibition of LID across the L-Dopa groups. Linear regression lines are plotted to best fit the data points. Stars represent statistically significant difference from baseline AIMS when L-Dopa and vehicle were administered. (n = 8 for each drug condition; unpaired two-tailed Student's t test * P<0.05, ** P<0.01. n.s. indicates P>0.05. Graph is plotted as mean +/- SEM.)

(I) EC50 data reveals a positive correlation between the SAG concentration needed for half maximal LID attenuation and the L-Dopa dose used to induce AIMS.

(B) Results:

(Line: 16 to 175): The severity of LID within each animal model of LID is dependent on the dose of L-Dopa administered [12]. We thus investigated whether the dose of Shh needed to attenuate AIMS might scale with the dose of L-Dopa used to induce dyskinesia in the $AK^{-/-}$ and 6-OHDA models. LID were induced in three groups of $AK^{-/-}$ mice injected daily for 20 days with either 5, 10 or 30 mg/kg L-Dopa (Fig. 1 G). In probe trials at days 21, 25, 29, and 33, the dose-dependent degree of LID attenuation by SAG was tested using a within-subject escalation strategy (0.8; 2.5; 7.5; and 15 mg/kg) in each L-Dopa dose group (5, 10 or 30 mg/kg; Fig. 1 G). While the lowest dose of SAG did not have a significant effect on AIMS in any of the L-Dopa dosing groups, the higher doses resulted in a dose dependent attenuation of AIMS in each of the L-dopa groups. This study revealed that the dose of SAG needed to attenuate LID by 50 % in each of the L-Dopa groups scaled with the dose of L-Dopa used to induce dyskinesia (Fig. 1 H and I).

10. Similar comments for $Smo(ChAT-cre^{-/-})$ mice where L-Dopa 25 mg/kg (again a huge dose) was used.

Author response: Guided by the dose response and EC50 study performed in the $AK^{-/-}$ paradigm we dosed all genetic models with 25 mg/kg L-Dopa. We reasoned that the genetic manipulation which takes hold during development (1) allows for more powerful and effective compensatory mechanisms than are available to the brain upon acute neurotoxic insult in the adult animal and (2) affect the same effector pathway (i.e. Smo signaling).

In order to facilitate a quick reference of all paradigm specific conditions we added a table as supplementary information that lists paradigms, conditions and assays performed.

Changes to manuscript:

(A) Figures:

Supplementary table 1 (see also responses above)

Supplementary table 2

(B) Results:

(Lines 232 to 238): To begin testing this, we produced mice with conditional ablation of *Smo* from cholinergic neurons (*Smo^{CIN}^{-/-}*, Fig. 4 A; generation of all conditional gene ablation and gain of function mouse lines described in methods; allelic configuration of experimental and control animals of all recombinant mouse lines used in this study is listed in supplementary table 2; all drug dosing regimens were started around 8 weeks of age and all final behavioral and histological analysis was carried out between 12 and 14 weeks of age).

- 6. MPTP-treated Macaques experiments. Only SAG molecule was tested in macaques, and one would expect also PUR testing. Also, it is important to provide the details of the statistics (exact p values) for these experiments Fig1F); Indeed, the information given in the Legend is not sufficient (p<0.05) to judge of the real stats since 4 animals have been used and the effect seems quite light and is not fully convincing. Moreover, in MPTP-treated macaque the DAN death rate is expected to be higher than in AK-/- mice (and in the range of those obtained with 6-OHDA-treated mice), and thus SAG 27 mg/kg is a very high dose. Again, it is important to strengthen these experiments to be able to firmly conclude about an effect of SAG.**

Author response: We thank the reviewer for pointing out the lack of specifics on the statistical analysis. The Friedman's non-parametric statistics factor (Fr) is equal to 10.89 resulting in a p value of 0.0020. Dunn post-hoc test (all doses compared to vehicle only) indicates that the 27 mg/kg dose is significantly lowering dyskinesia severity (p = 0.0185).

As indicated in the response to reviewer 1, attrition of anti-dyskinetic effects is a well-known phenomenon in the LID field [5] [6]. Manipulations which in rodents exhibit a large magnitude of effect are regularly decreased in non-human primate models and then sometimes further decreased in patients. We are encouraged nonetheless given that the effect remains significant despite an n of 4 and use of non-parametric statistical analysis which accord well with the latest power calculations in this field [33].

We agree with the reviewer about the need for a pharmacokinetic characterization of SAG in rodents and primates (e.g. Heine et al. asking for "determining the optimal dose schedule and further establish safety parameters (side effects, tumor development) in rodents and other animal systems (nonhuman primate) before clinical trials in humans" [34]) that knowingly require different dosages for reaching target engagement with the same drug. However, such data are surprisingly lacking despite SAG being a quite old molecule [7]. Searching through the literature, we have been unable to find classic pharmacokinetics/pharmacodynamics studies. No primary data are available to the best of our knowledge and most reviews focus upon the in vitro data (see also response to Reviewer #1, point 2) [8, 9]. For the rodent studies, we relied on the numerous empirical dose selections described in the literature. For the primate component, not knowing the pharmacokinetics or the target engagement dose of SAG, we adopted a careful dose-escalating regimen with veterinarian monitoring (on top of the video recordings) for occurrence of possible side-effects. A classic design would have been to select a full latin square design. However, a complete pharmacokinetic study in macaques for SAG and PUR in the context of LID is beyond the scope of this manuscript and the financial capability of this lab. We therefore chose a dose-escalation approach to give a first indication for potential translatability into primates.

Changes to manuscript:

(A) Results:

(Line 154 to 161): In macaques, treatment with the toxin 1-methyl-4-phenyl-1,2,3,6-tetrahydropyridine (MPTP) produces PD-like DAN degeneration. Chronic dosing with L-Dopa reduces parkinsonian disability among MPTP treated macaques but results in the establishment of LID that can be attenuated by amantadine [33]. We utilized subject-specific L-Dopa dosing regimens (18 to 22 mg/kg; see Methods) to establish AIMs in MPTP lesioned macaques. We then tested in a within-subject dose escalation experiment whether SAG administration (3; 9; 27 mg/kg) together with L-Dopa resulted in an attenuation of LID (Fig. 1 F). We found that the highest dose of SAG resulted in a modest but significant attenuation of LID (Fig. 1 F).

(B) Figures:

Figure 1 Legend:

Dyskinesia score in dyskinetic Macaques was reduced in response to a within-subject drug escalation of SAG at the highest dose (0, 3, 9, and 27 mg/kg; n=4; (Friedman's nonparametric RM one-way ANOVA, $F= 10.89$, $P= 0.0020$, followed by Dunn's post hoc test, $P= 0.0185$).

11. Pharmacology SAG and PUR. The authors conclude (Abstract and other section) that agonists (plural) of Smo alleviate LID, but PUR needs to be tested in AK^{-/-} mice (and not only in 6-OHDA-treated mice).

Optimally, PUR should have been tested in macaques to claim that Smo agonists are efficient in murine and non-human primates (as stated by the authors).

Authors Response: Agreed. We have removed all mention of agonist's "s" and instead have specifically stated that PUR was only tested in the 6-OHDA paradigm.

Changes to manuscript:

(A) Abstract:

(Line 47 – 49): We find that the pharmacological activation of Smoothed (Smo), a downstream effector of Shh, attenuates LID in the neurotoxic 6-OHDA- and genetic aphakia mouse models of PD.

(B) Results:

(Line 138 – 143): . A single dose of the Smo agonists Purmorphamine [35] (PUR, 15 mg/kg) or SAG (10 mg/kg) administered alongside the final dose of L-Dopa after repeated daily dosing with L-Dopa alone significantly reduced AIMs in the unilateral 6-OHDA model of LID (Fig. 1 D, blue bars). A single dose of SAG (20 mg/kg) in the AK^{-/-} model of LID following repeated L-Dopa administration alone also reduced AIMs significantly (Fig. 1 E, blue bar).

12. Result part, second chapter. The authors used SAG 25 mg/kg in Smo(ChAT-cre^{-/-}) mice. Why such dose since in Figure 1 there used 20 mg/kg?

Authors response: We did not dose *Smo^{CIN}^{-/-}* or *SmoM2^{CIN}^{+/-}* mice with SAG in the context of measuring AIMs. There is no such mention in the second chapter. Since Smo is the target of SAG no effect would be expected in mice without Smo as demonstrated in Figure 7 C and D. Since SmoM2 is constitutively active, SAG is not expected to increase SmoM2 activity.

We added two supplementary tables listing all paradigms, conditions, and assays performed for quick reference.

Changes to manuscript:

(A) Figures:

Supplementary table 1

Supplementary table 2

Reviewer #3: Minor comments:

1) Abstract: add “Smo” abbreviation when it is first cited.

Authors Response: Done.

2) Results text: Fig. 1B and C, there is no “white bars” but grey bars.

Authors Response:

Text changed in results from “white” to “grey”; text changed in figure legend from “white” to “grey”.

3) Results: DAN are unable to follow (fire) optogenetic activation at 60 Hz. Why did you choose to apply 60 Hz stimulation?

Authors Response: We wanted to achieve forced Shh release not physiological DAN firing. We were guided by reports that 60 Hz burst stimulation in the Hippocampus resulted in Shh release [36]. Our bidirectional titration experiments reveal that Shh becomes limiting over time under these conditions. We assume that even if DAN cannot follow 60 Hz, the optogenetic stimulation will result in increased activity of DAN.

1. Bastide, M.F., et al., *Pathophysiology of L-dopa-induced motor and non-motor complications in Parkinson's disease*. Progress in neurobiology, 2015. **132**: p. 96-168.
2. Sebastianutto, I., et al., *Validation of an improved scale for rating L-DOPA-induced dyskinesia in the mouse and effects of specific dopamine receptor antagonists*. Neurobiology of disease, 2016. **96**: p. 156-170.
3. Lundblad, M., et al., *A model of L-DOPA-induced dyskinesia in 6-hydroxydopamine lesioned mice: relation to motor and cellular parameters of nigrostriatal function*. Neurobiology of disease, 2004. **16**(1): p. 110-23.
4. Ding, Y., et al., *Chronic 3,4-dihydroxyphenylalanine treatment induces dyskinesia in aphakia mice, a novel genetic model of Parkinson's disease*. Neurobiology of disease, 2007. **27**(1): p. 11-23.
5. Bastide, M.F., et al., *Pathophysiology of L-dopa-induced motor and non-motor complications in Parkinson's disease*. Prog Neurobiol, 2015. **132**: p. 96-168.
6. Fox, S.H. and J.M. Brotchie, *Viewpoint: Developing drugs for levodopa-induced dyskinesia in PD: Lessons learnt, what does the future hold?* Eur J Neurosci, 2019. **49**(3): p. 399-409.
7. Chen, J.K., et al., *Small molecule modulation of Smoothed activity*. Proc Natl Acad Sci U S A, 2002. **99**(22): p. 14071-6.
8. Hadden, M.K., *Hedgehog pathway agonism: therapeutic potential and small-molecule development*. ChemMedChem, 2014. **9**(1): p. 27-37.
9. Stanton, B.Z. and L.F. Peng, *Small-molecule modulators of the Sonic Hedgehog signaling pathway*. Mol Biosyst, 2010. **6**(1): p. 44-54.
10. Fox, S.H., et al., *A critique of available scales and presentation of the Non-Human Primate Dyskinesia Rating Scale*. Mov Disord, 2012. **27**(11): p. 1373-8.

11. Gonzalez-Reyes, L.E., et al., *Sonic hedgehog maintains cellular and neurochemical homeostasis in the adult nigrostriatal circuit*. *Neuron*, 2012. **75**(2): p. 306-19.
12. Arensdorf, A.M., et al., *Sonic Hedgehog Activates Phospholipase A2 to Enhance Smoothed Ciliary Translocation*. *Cell Rep*, 2017. **19**(10): p. 2074-2087.
13. Arensdorf, A.M., S. Marada, and S.K. Ogden, *Smoothed Regulation: A Tale of Two Signals*. *Trends Pharmacol Sci*, 2016. **37**(1): p. 62-72.
14. Ogden, S.K., et al., *G protein G α functions immediately downstream of Smoothed in Hedgehog signalling*. *Nature*, 2008. **456**(7224): p. 967-70.
15. Riobo, N.A., et al., *Activation of heterotrimeric G proteins by Smoothed*. *Proc Natl Acad Sci U S A*, 2006. **103**(33): p. 12607-12.
16. Dessaud, E., A.P. McMahon, and J. Briscoe, *Pattern formation in the vertebrate neural tube: a sonic hedgehog morphogen-regulated transcriptional network*. *Development*, 2008. **135**(15): p. 2489-503.
17. Yan, Z. and D.J. Surmeier, *Muscarinic (m2/m4) receptors reduce N- and P-type Ca²⁺ currents in rat neostriatal cholinergic interneurons through a fast, membrane-delimited, G-protein pathway*. *The Journal of neuroscience : the official journal of the Society for Neuroscience*, 1996. **16**(8): p. 2592-604.
18. Yan, Z., W.J. Song, and J. Surmeier, *D2 dopamine receptors reduce N-type Ca²⁺ currents in rat neostriatal cholinergic interneurons through a membrane-delimited, protein-kinase-C-insensitive pathway*. *J Neurophysiol*, 1997. **77**(2): p. 1003-15.
19. Goldberg, J.A. and C.J. Wilson, *Control of spontaneous firing patterns by the selective coupling of calcium currents to calcium-activated potassium currents in striatal cholinergic interneurons*. *J Neurosci*, 2005. **25**(44): p. 10230-8.
20. Choi, S.J., et al., *Alterations in the intrinsic properties of striatal cholinergic interneurons after dopamine lesion and chronic L-DOPA*. *Elife*, 2020. **9**.
21. Shen, W., et al., *M4 Muscarinic Receptor Signaling Ameliorates Striatal Plasticity Deficits in Models of L-DOPA-Induced Dyskinesia*. *Neuron*, 2016. **90**(5): p. 1139.
22. Brugnoli, A., C.A. Pisano, and M. Morari, *Striatal and nigral muscarinic type 1 and type 4 receptors modulate levodopa-induced dyskinesia and striato-nigral pathway activation in 6-hydroxydopamine hemilesioned rats*. *Neurobiol Dis*, 2020. **144**: p. 105044.
23. Bordia, T., et al., *Optogenetic activation of striatal cholinergic interneurons regulates L-dopa-induced dyskinesias*. *Neurobiology of disease*, 2016. **91**: p. 47-58.
24. Quik, M., et al., *Nicotine-mediated improvement in L-dopa-induced dyskinesias in MPTP-lesioned monkeys is dependent on dopamine nerve terminal function*. *Neurobiology of disease*, 2013. **50**: p. 30-41.
25. Quik, M., et al., *Nicotine reduces established levodopa-induced dyskinesias in a monkey model of Parkinson's disease*. *Movement disorders : official journal of the Movement Disorder Society*, 2013. **28**(10): p. 1398-406.
26. Won, L., et al., *Striatal cholinergic cell ablation attenuates L-DOPA induced dyskinesia in Parkinsonian mice*. *The Journal of neuroscience : the official journal of the Society for Neuroscience*, 2014. **34**(8): p. 3090-4.
27. Girasole, A.E., et al., *A Subpopulation of Striatal Neurons Mediates Levodopa-Induced Dyskinesia*. *Neuron*, 2018. **97**(4): p. 787-795 e6.
28. Fasano, S., et al., *Inhibition of Ras-guanine nucleotide-releasing factor 1 (Ras-GRF1) signaling in the striatum reverts motor symptoms associated with L-dopa-induced dyskinesia*. *Proc Natl Acad Sci U S A*, 2010. **107**(50): p. 21824-9.

29. Ding, Y., et al., *Enhanced striatal cholinergic neuronal activity mediates L-DOPA-induced dyskinesia in parkinsonian mice*. Proceedings of the National Academy of Sciences of the United States of America, 2011. **108**(2): p. 840-5.
30. Pahwa, R., et al., *Amantadine extended release for levodopa-induced dyskinesia in Parkinson's disease (EASED Study)*. Movement disorders : official journal of the Movement Disorder Society, 2015. **30**(6): p. 788-95.
31. Nunes, I., et al., *Pitx3 is required for development of substantia nigra dopaminergic neurons*. Proc Natl Acad Sci U S A, 2003. **100**(7): p. 4245-50.
32. Hwang, D.Y., et al., *Selective loss of dopaminergic neurons in the substantia nigra of Pitx3-deficient aphakia mice*. Brain Res Mol Brain Res, 2003. **114**(2): p. 123-31.
33. Stanley, P., et al., *Meta-analysis of amantadine efficacy for improving preclinical research reliability*. Mov Disord, 2018. **33**(10): p. 1555-1557.
34. Heine, V.M., et al., *A small-molecule smoothed agonist prevents glucocorticoid-induced neonatal cerebellar injury*. Sci Transl Med, 2011. **3**(105): p. 105ra104.
35. Sinha, S. and J.K. Chen, *Purmorphamine activates the Hedgehog pathway by targeting Smoothed*. Nat Chem Biol, 2006. **2**(1): p. 29-30.
36. Su, Y., et al., *High frequency stimulation induces sonic hedgehog release from hippocampal neurons*. Sci Rep, 2017. **7**: p. 43865.

REVIEWERS' COMMENTS:

Reviewer #1 (Remarks to the Author):

The authors have very thoughtfully responded to each of the issues raised and the manuscript should be published without delay. Well done!

Reviewer #2 (Remarks to the Author):

In this revised version of the manuscript, the authors replied to all my queries. Overall the current version of the manuscript has significantly improved, and represents novel and substantial scientific evidence which deserves full consideration.

Reviewer #3 (Remarks to the Author):

The authors have answered satisfactorily to most of my comments.